# Differentiable Generalized Sliced Wasserstein Plans

**Laetitia Chapel**[*]
IRISA – L'Institut Agro Rennes-Angers
Rennes, France.
`laetitia.chapel@irisa.fr`

**Romain Tavenard**[*]
Université de Rennes 2 – IRISA
Rennes, France.
`romain.tavenard@univ-rennes2.fr`

**Samuel Vaiter**
CNRS & Université Côte d'Azur
Laboratoire J. A. Dieudonné
Nice, France.
`samuel.vaiter@cnrs.fr`

## Abstract

Optimal Transport (OT) has attracted significant interest in the machine learning community, not only for its ability to define meaningful distances between probability distributions – such as the Wasserstein distance – but also for its formulation of OT plans. Its computational complexity remains a bottleneck, though, and slicing techniques have been developed to scale OT to large datasets. Recently, a novel slicing scheme, dubbed min-SWGG, lifts a single one-dimensional plan back to the original multidimensional space, finally selecting the slice that yields the lowest Wasserstein distance as an approximation of the full OT plan. Despite its computational and theoretical advantages, min-SWGG inherits typical limitations of slicing methods: (i) the number of required slices grows exponentially with the data dimension, and (ii) it is constrained to linear projections. Here, we reformulate min-SWGG as a bilevel optimization problem and propose a differentiable approximation scheme to efficiently identify the optimal slice, even in high-dimensional settings. We furthermore define its generalized extension for accommodating to data living on manifolds. Finally, we demonstrate the practical value of our approach in various applications, including gradient flows on manifolds and high-dimensional spaces, as well as a novel sliced OT-based conditional flow matching for image generation – where fast computation of transport plans is essential.

## 1 Introduction

Optimal Transport (OT) has emerged as a foundational tool in modern machine learning, primarily due to its capacity to provide meaningful comparisons between probability distributions. Rooted in the seminal works of Monge [38] and Kantorovich [27], OT introduces a mathematically rigorous framework that defines distances, such as the Wasserstein distance, that have demonstrated high performance in various learning tasks. Used as a loss function, it is now the workhorse of learning problems ranging from classification, transfer learning or generative modelling, see [39] for a review. One of the key advantages of OT lies in its dual nature: it also constructs an optimal coupling or transport plan between distributions. This coupling reveals explicit correspondences between samples, enabling a wide range of applications. For example, OT has proven valuable in shape matching [10], color transfer [47, 49], domain adaptation [14], and, more recently, in generative modeling through conditional flow matching [45, 54], where it provides an alignment between the data distribution and samples from a prior.

---

[*]equal contribution.

39th Conference on Neural Information Processing Systems (NeurIPS 2025).

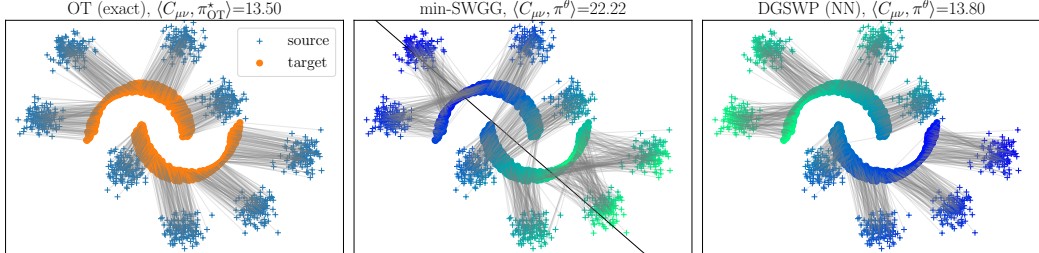

Figure 1: 8Gaussians (source) to Two Moons (target) distributions and associated OT plans in Grey: (Left) exact solution (Middle) min-SWGG, that projects samples on an optimal line determined by random sampling (Right) Differentiable Generalized SW plan, that relies on a neural network to get non linear-based ordering of the samples. Gradient of colors represent the ordering of the samples.

Despite the many successes of optimal transport in machine learning, computing OT plans remains a computationally challenging problem. The most used exact algorithms are drawn from linear programming, typically resulting in a $O(n^3)$ complexity with respect to the number of samples $n$. To alleviate this issue, several strategies have been developed in the last decade, that can be roughly classified in three families: *i)* regularization-based methods, such as entropic optimal transport [16], *ii)* minibatch-based methods [23, 20], that average the outputs of several smaller optimal transport problems and *iii)* approximation-based through closed-form formulas such as projection-based OT. This paper is concerned with this third class of methods, with the underlying goal to obtain a transport plan. We quickly review them below.

**Approximation of OT loss with sliced-OT.** The Sliced-Wasserstein Distance (SWD) [48, 11] approximates the Wasserstein distance by projecting onto one-dimensional subspaces, where OT has a closed-form solution obtained in $O(n \log n)$. While SWD averages over multiple projections, max-SWD [18] selects the most informative one, enabling efficient computations on large-scale problems while preserving key properties. Generalized SWD alleviates the inefficiencies caused by the linear projections using polynomial projections or neural networks [30]. They build on generalized Radon transforms to use polynomial projections or neural networks, and provide conditions for which it remains a valid distance: the generalized Radon transform must be injective. Rather than considering non-linear projections, Chen et al. [12] aim at better capturing non-linearities by *augmenting* the input space, using injective neural networks, such that linear projections could better capture them. Generalized SWD has also been defined in the case of tree metrics [55].

**Approximation of OT plan with sliced-OT.** Addressing the challenge that none of the previous works provide an approximated transport plan, [37] proposes min-SWGG, a slicing scheme that lifts one-dimensional OT plans to the original space; [35] follow the same line but rather define expected plans computed over all the lines instead of retaining the best one.

**Differentiable OT.** As it is a discrete problem, OT is not differentiable *per se*. Several formulations, that mostly rely on *smoothing* the value function by adding a regularization term, have then been defined, the entropic-regularized version of the OT problem [16] being one of the most striking examples. It can be efficiently solved via Sinkhorn's matrix scaling algorithm, providing an OT approximation that is differentiable. Blondel et al. [7] use a $\ell_2$ regularization term to define a smooth and sparse approximation. Regarding the unregularized OT formulation, an approximated gradient can be computed using sub-gradient on the dual formulation (see [22] for instance).

**Optimizing on discrete problems.** The OT problem is a linear program and the optimal plan belong to the (potentially rescaled) Birkhoff polytope, *i.e.* the set of doubly-stochastic matrices, which is discrete. It is also the case for a wide range of problems for which the solution is prescribed to belong to such a discrete set. For instance, one can cite the sorting operator, the shortest path or the top-$k$ operator. They break back-propagation along the computational graph, hence preventing their use in deep learning pipelines. Numerous works have been proposed to provide differentiable proxies, based on smoothing the operators: dataSP [32] define a differentiable all-to-all shortest path algorithm, smoothed sorting and ranking [8] can be defined by projection onto the permutahedron, perturbed optimizers [4] have also been defined, relying on perturbing the input data, to name a few.

**Contributions.** In this paper, we aim at approximating the OT distance and plan. We rely on a slicing scheme, extending the framework of [37], highlighting that it is an instance of a bilevel optimisation problem. We provide two main contributions: a generalized Sliced-Wasserstein that provides approximated OT plans relying on non-linear projections (see Fig. 1 for an illustration), then a GPU-friendly optimization algorithm to find the best projector. We showcase the benefits brought by these contributions in three different experimental contexts: on a 2 dimensional example where a non-linear projection is sought; conducting gradient flow experiment in high dimensional space; introducing a novel sliced OT-based conditional flow matching for image generation.

## 2 Preliminaries on Optimal Transport

We now give the necessary background on Optimal Transport and on sliced-based approximations. For a reference on computational OT, the reader can refer to [44].

### 2.1 Discrete Optimal Transport

We consider two points clouds $\{x_i\}_{i=1}^n \in \mathcal{X}^n$ and $\{y_i\}_{i=1}^m \in \mathcal{X}^m$ where $\mathcal{X} \subset \mathbb{R}^d$ is a discrete subset. We denote their associated empirical distributions $\mu = \sum_{i=1}^n a_i \delta_{x_i}$ and $\nu = \sum_{i=1}^m b_i \delta_{y_i}$.

**Formulation and Wasserstein distance.** We consider a cost function $c$ that will be in our setting $c(u, v) = \|u - v\|_p^p$ for $p > 1$ and $C_{\mu\nu} = (c(x_i, y_j))_{i=1, j=1}^{n,m}$. Traditional Kantorovich formulation of optimal transport is given by the Wasserstein metric on $\mathcal{P}(\mathbb{R}^d)$ defined as

$$W_p^p(\mu, \nu) = \min_\pi \sum_{i=1}^n \sum_{j=1}^m \pi_{i,j} \|x_i - y_j\|_p^p \quad \text{subject to} \quad \pi \in U(a, b), \tag{1}$$

where $U(a, b) = \{\pi \in \mathbb{R}_{\geq 0}^{n \times m} : \pi 1_m = a, \pi^\top 1_n = b\}$ is the set of couplings between $\mu$ and $\nu$, Eq. (1) is a convex optimization problem and an optimal transport plan $\pi_{\text{OT}}^\star$ is a solution. Note that, *a priori*, there is no reason for this minimizer to be uniquely defined.

**One-dimensional OT.** When $d = 1$, and $\mu, \nu$ are empirical distributions with $a_i = b_i = 1/n$, the optimal transport problem is equivalent to the assignment problem. In this case, the Wasserstein distance can be computed by sorting the empirical samples, resulting in an overall complexity of $O(n \log n)$. Let $\sigma$ and $\tau$ be permutations such that $x_{\sigma(1)} \leq x_{\sigma(2)} \leq \ldots \leq x_{\sigma(n)}$ and $y_{\tau(1)} \leq y_{\tau(2)} \leq \ldots \leq y_{\tau(n)}$. The Wasserstein distance is then given by $W_p^p(\mu, \nu) = \frac{1}{n} \sum_{i=1}^n |x_{\sigma(i)} - y_{\tau(i)}|^p$. The optimal transport is thus monotone and its plan $\pi_{\text{OT}}^\star$ has the form of a permutation matrix. Note that this approach can be easily extended to $n \neq m$ and arbitrary marginals $a, b$ [44].

### 2.2 Sliced Wasserstein Distance

**Formulation.** Sliced-Wasserstein (SWD) relies on a simple idea: disintegrate the original problem onto unidimensional ones, and average over the different solutions. More precisely, SWD [48] approximates the Wasserstein distance by averaging along projection directions $\theta \in \mathbb{S}^{d-1}$ as

$$SWD_p^p(\mu, \nu) := \int_{\mathbb{S}^{d-1}} W_p^p(P_\sharp^\theta \mu, P_\sharp^\theta \nu) \mathrm{d}\lambda(\theta), \tag{2}$$

where $P^\theta : \mathbb{R}^d \to \mathbb{R}$ is the 1D projection onto the unit vector $\theta$, $P^\theta(x) = \langle x, \theta \rangle$, and $\lambda$ is the uniform distribution on the unit sphere $\mathbb{S}^{d-1}$. Typically, SWD is computed thanks to a Monte-Carlo approximation in which $L$ directions are drawn independently, leading to a computational complexity of $O(dLn + Ln \log n)$. One of the main drawback of SWD is that is requires a high number of random projections, which leads to intractability for high dimensional problems. Then, there have been works to perform selective sampling, *e.g.* [41], or to optimize over the directions [18, 37]. The two later works rely on non-convex formulations that can be optimized.

**Generalized Sliced WD.** The idea of non-linear slicing has been explored in several works, with the aim to improve the projection efficiency, *e.g.* when the data live on non-linear manifolds. In [30], the generalized SWD uses nonlinear projections such as neural network-based ones; the conditions on which it yields a valid metric are also stated: the projection map must be injective. In [41], a generalized projection is also proposed in addition to selective sampling. Augmented Sliced Wasserstein

(AWD) [12] pursued the same goal but, rather than considering nonlinear projections, they use a neural network to *augment* the input space, which leads to a space on which a linear projection better fits the data. All these works lead to significant improvements in a wide set of machine learning scenarios.

## 2.3 Sliced Wasserstein Plan

As it is defined as an average over 1D OT distances, SWD does not provide inherently a transport plan[2]. Recently, some works have tackled this limitation by lifting one or several one-dimensional plans back to the original multidimensional space [37, 35]. We now give more details on [37] as [35] extend the latter framework to by averaging several plans.

**Sliced Wasserstein Generalized Geodesics.** Mahey et al. [37] introduced an OT surrogate that lifts the plan from the 1D projection onto the original space. In more details, when $n = m$, it is defined as:

$$\text{min-SWGG}_p^p(\mu, \nu) = \min_{\theta \in \mathbb{S}^{d-1}} \text{SWGG}_p^p(\mu, \nu, \theta) := \frac{1}{n} \sum_{i=1}^{n} \|x_{\sigma_\theta(i)} - y_{\tau_\theta(i)}\|_p^p \qquad (3)$$

where $\sigma_\theta$ and $\tau_\theta$ are the permutations obtained by sorting $P_\sharp^\theta \mu$ and $P_\sharp^\theta \nu$. Note that it extends naturally to the case where $n \neq m$ which we omit here for brevity. By setting $p = 2$, it hinges on the notion of Wasserstein generalized geodesics [2] with pivot measure supported on a line. This alternative formulation allows deriving an optimization scheme to find the optimal $\theta$ that relies on multiple *copies* of the projected samples, which holds only for $p = 2$. For a fixed direction $\theta$, provided that families $(P_\sharp^\theta(x_i))_i$ and $(P_\sharp^\theta(y_i))_i$ are injective, that is to say there is no ambiguity on the orderings $\sigma_\theta$ and $\tau_\theta$, $\text{SWGG}(\cdot, \cdot, \theta)$ is itself a metric [36]. Min-SWGG has appealing properties: it yields an upper bound of the Wasserstein distance that still provides an explicit transport map between the input measures. The authors show that min-SWGG metrises weak convergence and is translation-equivariant.

**Limitations of min-SWGG.** While min-SWGG provides an upper bound on the Wasserstein distance, its tightness is not guaranteed in general. However, two settings are known where the bound is exact: (i) when one of the distributions is supported on a one-dimensional subspace ; and (ii) when the ambient dimension satisfies $d \geq 2n$ [37]. More generally, the number of reachable permutations increases with the ambient dimension [15], making data dimensionality a critical factor in the quality of the approximation. These insights motivate our first contribution: the definition of Generalized Wasserstein plans, which aim to extend the set of reachable permutations and better capture non-linear structures.

## 3 Generalized Sliced Wasserstein Plan

Generalized Sliced Wasserstein Plan (GSWP) is built upon the idea of generalizing SWGG to an arbitrary scalar field parameterized by some $\theta \in \mathbb{R}^q$. We consider a map $\phi : \mathbb{R}^d \times \mathbb{R}^q \to \mathbb{R}$, typically a one-dimensional projection $\phi(x, \theta) = \langle x, \theta \rangle$ (in which case $q = d$), or a neural network, as in Fig. 1 for example. A Generalized Sliced Wasserstein Plan distance is defined as the one-dimensional Wasserstein distance between the point clouds through the image of $\phi^\theta := \phi(\cdot, \theta)$ for a given $\theta \in \mathbb{R}^q$.

**Definition 1.** Let $p > 1$, $\theta \in \mathbb{R}^q$ and $\mu, \nu \in \mathcal{P}(\mathcal{X})$. The $\theta$-Generalized Sliced Wasserstein Plan distance between $\mu$ and $\nu$ is defined as

$$d^\theta(\mu, \nu) = W_p(\phi_\sharp^\theta \mu, \phi_\sharp^\theta \nu).$$

Any element $\pi^\theta(\mu, \nu) \in U(a, b)$ that achieves $d^\theta(\mu, \nu)$ is called a $\theta$-Generalized Sliced Wasserstein Plan ($\theta$-GSWP).

Denoting $\theta \mapsto C_{\mu\nu}^\phi(\theta)$ the cost matrix between $\mu, \nu$ through the image of $\phi$, a $\theta$-GSWP reads

$$\pi^\theta \in \arg\min_{\pi \in U(a, b)} g(\pi, \theta) := \langle C_{\mu\nu}^\phi(\theta), \pi \rangle = \sum_{i=1}^{n} \sum_{j=1}^{m} \pi_{i,j} |\phi(x_i, \theta) - \phi(y_j, \theta)|^p. \qquad (4)$$

The $g(x, \cdot)$ function is continuous but not convex, as illustrated in Fig. 2. Note that, *i)* for $p > 1$, the map $C_{\mu\nu}^\phi$ has the same regularity as $\phi$ and *ii)* a solution $\pi^\theta$ is a suboptimal point of the standard (1) problem. Indeed, since the optimization occurs on the same coupling space $U(a, b)$, $\pi^\theta$ is an admissible point for the original problem. GSWP defines a distance on the space of measures $\mathcal{P}(\mathcal{X})$.

---

[2]Note that a transport plan can also be inferred when performing SW gradient flows by putting into correspondence the original and final samples when the algorithm has converged.

**Proposition 1.** *Let $\theta \in \mathbb{R}^q$ and assume $\phi^\theta$ is an injective map on $\mathcal{X}$. Then $d^\theta$ is a distance on $\mathcal{P}(\mathcal{X})$.*

For clarity, all our proofs are presented in Sec A.1. Note that the injectivity condition is akin to designing sufficient and necessary conditions for the injectivity of generalized Radon transform [5] and relates to the problem of reconstructing measures from discrete measurements [52]. In practice, using a general $\phi$ increases expressivity, enabling a wider range of permutations compared to a linear map. By an application of Rockafellar's enveloppe theorem, we also have a characterization of the gradient of $d^\theta$ as a function of $\theta$. Let us assume that $p > 1$, $\phi$ is jointly $C^1$ and (4) has a unique solution at $\theta \in \mathbb{R}^q$. Then we have $\nabla_\theta d^\theta = \frac{\partial C_{\mu\nu}^\phi}{\partial \theta}(\theta)^\top \pi^\theta$. Note that a similar result on the subdifferential of $d^\theta$ is true if $\phi$ is not differentiable but convex thanks to Danskin's theorem.

**Definition 2.** Let $p \geq 0$ and $\mu, \nu \in \mathcal{P}(\mathcal{X})$. The minimal Generalized Sliced Wasserstein Plan semimetric min-GSWP between $\mu$ and $\nu$ is defined as

$$\text{min-GSWP}_p^p(\mu, \nu) = \min_{\theta \in \mathbb{R}^q} h(\theta) := f(\pi^\theta) := \langle C_{\mu\nu}, \pi^\theta \rangle \tag{5}$$

$$\text{subject to} \quad \pi^\theta \in \arg \min_{\pi \in U(a,b)} g(\pi, \theta).$$

Equation (5) defines a bilevel optimization problem. In the case where $\phi(x, \theta) = \langle x, \theta \rangle$ and if we further constrain $\theta$ to live on the unit sphere, (5) is the min-SWGG [37] approximation of OT. Observe that since $\pi^\theta$ is an admissible coupling for the original OT problem (1), min-GSWP$(\mu, \nu)$ is an upper bound for $W_p(\mu, \nu)$. Unfortunately, the value function $\theta \mapsto h(\theta)$ does not, in general, possess desirable regularity properties, even for the simple choice of a one-dimensional projection $\phi(x, \theta) = \langle x, \theta \rangle$: it is discontinuous, as illustrated in Fig. 2. Moreover, as soon as the maps $\phi(x, \theta)$ and $\phi(y, \theta)$ give rise to the same permutations $\sigma_\theta$ and $\tau_\theta$, the value of $\pi^\theta$ remains the same. As a consequence, one cannot use (stochastic) gradient-based bilevel methods such as [43, 17, 3] on (5).

It turns out that even if min-SWGG was introduced as the minimization over the sphere $\mathbb{S}^{d-1}$, it is possible to see it as an unconstrained optimization problem thanks to the following lemma.

**Lemma 1.** *Assume that $\theta \mapsto \phi(x, \theta)$ is 1-homogeneous for all $x \in \mathbb{R}^d$. Then, $\theta \mapsto d^\theta(\mu, \nu)$ is 1-homogeneous, and $h$ is invariant by scaling: $h(c\theta) = h(\theta)$ for all $c > 0$.*

In particular, for every open set where $h$ is differentiable, if $\phi$ is 1-homogeneous, the gradient flow $\dot{\theta} = -\nabla h(\theta)$ has orthogonal level lines $\langle \dot{\theta}, \theta \rangle = 0$, hence the dynamics occurs on the sphere of radius equal to the norm of the initialization. In practice, it means – and we observed – that we do not have to care about the normalization of $\theta$ during our gradient descent.

When $\phi(x, \theta) = \langle x, \theta \rangle$, min-SWGG can be smoothed by making *perturbed* copies of the projections $\phi(x, \theta)$; heuristics for determining the number of copies and the scale of the noise are given in [37]. Nevertheless, the continuity of the obtained smooth surrogate cannot be guaranteed, and the formulation is only valid for $p = 2$. The following section proposes a differentiable approximation of min-GSWP rooted in probability theory which is valid for any $p > 1$. The main difference with the aforementioned scheme is that we here perturb the parameters (or direction) $\theta$ rather than the samples.

## 4 Differentiable Approximation of min-GSWP

Smoothing estimators by averaging is a popular way to tackle nonsmooth problems. We mention three families of strategies: *i)* smoothing by (infimal) convolution, *ii)* smoothing by using a Gumbel-like trick [1, 4], and *iii)* smoothing by reparameterization, and in particular by using Stein's lemma. Strategy *i)* would be computationally intractable in high dimension and strategy *ii)* doesn't yield a consistent transport plan due to perturbations in the cost matrix. We then focus on the third option.

We begin by recalling this classical result due to Stein [51], which plays a central role in our analysis. The lemma, restated below in our specific setting, provides an identity for computing gradients through expectations involving Gaussian perturbations. Typically, Stein's lemma is stated for "almost differentiable" function, we require here slightly less regularity

**Assumption 1.** There exists a open set $C \subseteq \mathbb{R}^q$ with Hausdorff dimension $\mathcal{H}^{q-1}(\mathbb{R}^q \setminus C) = 0$ (in particular Lebesgue-negligible) such that the mapping $\theta \mapsto h(\theta)$ is continuously differentiable on $C$.

Under this assumption, Stein's lemma remains true.

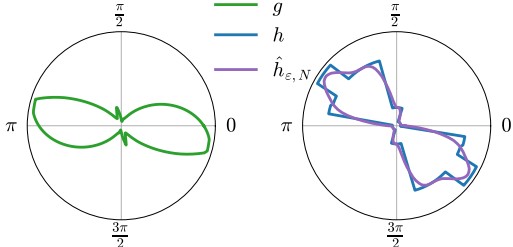

Figure 2: Example $g$ and $h$ (seen as a function of $\theta$ on the sphere $\mathbb{S}^1$) for a 2D OT problem. $\hat{h}_{\varepsilon,N}$ is a Monte-Carlo estimate of $h_\varepsilon$ with gradient $\hat{\nabla} h_{\varepsilon,N}$. Note that $g$ is continuous whereas $h$ is piecewise constant, hence the need for a smoothing mechanism, that results in $\hat{h}_{\varepsilon,N}$.

**Lemma 2** (Stein's lemma). *Suppose Assumption 1 holds and that for all indices $j$, the partial derivatives satisfy the integrability condition $\mathbb{E}[|\partial_j h(Z)|] < +\infty$. Let $Z \sim \mathcal{N}(0, \mathrm{Id}_q)$ be a standard multivariate Gaussian random variable, and $\varepsilon > 0$. Then, the following identity holds:*

$$\mathbb{E}_Z[\nabla h(\theta + \varepsilon Z)] = \varepsilon^{-1}\mathbb{E}_Z[h(\theta + \varepsilon Z)Z].$$

Note that this version of Stein's lemma is slightly more general than the original of Stein [51] in term of regularity asked to the value function $h$ (in particular, we do not require it to be absolutely continuous).

We define the following smoothed value function $h_\varepsilon$, which corresponds to a Gaussian smoothing of the original non-differentiable outer optimization problem:

$$h_\varepsilon(\theta) = \mathbb{E}_Z\left[h(\theta + \varepsilon Z)\right] = \langle C_{\mu\nu}, \pi_\varepsilon^\theta \rangle$$

where the $\theta$-**Differentiable Generalized Sliced Wasserstein Plan** ($\theta$-**DGSWP**) at smoothing level $\varepsilon$ reads

$$\pi_\varepsilon^\theta = \mathbb{E}_Z\left[\arg\min_{\pi \in U(a,b)} g(\pi, \theta + \varepsilon Z)\right]. \tag{6}$$

Due to the nice regularity properties of Stein's approximation, we get the following proposition regarding our approximation of $\theta$-GSWP.

**Proposition 2.** *Suppose Assumption 1 holds. Let $\mu, \nu \in \mathcal{P}(\mathcal{X})$. The following statements are true:*

1. *(Admissibility.) For any $\varepsilon > 0$, $\pi_\varepsilon^\theta$ is admissible (i.e., $\pi_\varepsilon^\theta \in U(a,b)$). Hence, $h_\varepsilon(\theta)$ gives an upper-bound of the Wasserstein metric $h_\varepsilon(\theta) \geq W_p^p(\mu,\nu)$.*

2. *(Differentiability.) For any $\varepsilon > 0$, the map $\theta \mapsto h_\varepsilon(\theta)$ is differentiable. Moreover, we have*
$$\nabla_\theta h_\varepsilon(\theta) = \varepsilon^{-1}\mathbb{E}_Z[h(\theta + \varepsilon Z)Z].$$

3. *(Consistency.) For almost all $\theta \in \mathbb{R}^q$, $\lim_{\varepsilon \to 0} h_\varepsilon(\theta) = h(\theta)$, and if $\nabla h(\theta)$ exists, then $\lim_{\varepsilon \to 0} \nabla h_\varepsilon(\theta) = \nabla h(\theta)$*

4. *(Distance.) Let $\theta \in \mathbb{R}^q$, if $\phi^\theta$ is injective on $\mathcal{X}$ then the map $(\mu, \nu) \mapsto (h(\theta)(\mu, \nu))^{1/p}$ is a distance on $\mathcal{P}(\mathcal{X})$. Assume that for almost all $\theta \in \mathbb{R}^q$, $\phi^\theta$ is injective. Then, $(\mu, \nu) \mapsto (h_\varepsilon(\theta)(\mu, \nu))^{1/p}$ is also a distance on $\mathcal{P}(\mathcal{X})$.*

While this result allows for the unbiased estimation of gradients, it is known that a naive Monte Carlo approximation of the right-hand side tends to suffer from high variance. To address this, one may consider an alternative formulation that often results in a reduced variance estimator. Specifically, using a control variate approach [6], one observes that

$$\mathbb{E}_Z[\nabla h(\theta + \varepsilon Z)] = \varepsilon^{-1}\mathbb{E}_Z[(h(\theta + \varepsilon Z) - h(\theta))Z], \tag{7}$$

which holds due to the zero mean of the Gaussian distribution and the linearity of the expectation. Equation (7) leads to a Monte-Carlo estimator of the gradient $\nabla h_\varepsilon(\theta)$ defined as

$$\hat{\nabla} h_{\varepsilon,N}(\theta) = \frac{1}{\varepsilon N}\sum_{k=1}^N (h(\theta + \varepsilon z_k) - h(\theta))z_k = \frac{1}{\varepsilon N}\sum_{k=1}^N \langle C_{\mu\nu}, \pi^{\theta + \varepsilon z_k} - \pi^\theta \rangle z_k, \tag{8}$$

where the vectors $z_i \sim \mathcal{N}(0, \mathrm{Id}_q)$ are independent standard Gaussian samples. Hence, estimating the Monte-Carlo gradient $\nabla h_{\varepsilon,N}(\pi, \theta)$ requires to solve $N + 1$ 1D-optimal transport problems for an overall cost of $O(N(n+m)\log(n+m))$. Algorithm 1 (in Supplementary material) describes a gradient descent method to perform the minimization of $h_\varepsilon$ using this Monte-Carlo approximation.

# 5 Experiments

We evaluate the performance of our Differentiable Generalized Sliced Wasserstein Plans, coined DGSWP, by assessing its ability to provide a meaningful approximated OT plan in several contexts. First, we consider a toy example where a non-linear projection must be considered; we then perform gradient flow experiments on Euclidean and hyperbolic spaces, demonstrating the versatility of our approach. Finally, we integrate sliced-OT plans in an OT-based conditional flow matching in lieu of mini-batch OT. In all the experiments, we use $\varepsilon = 0.05$ and $N = 20$ as DGSWP-specific hyperparameters. Full experimental setups and additional results are provided in App. A.3. Implementation is available online[3]; we also use POT toolbox [21].

## 5.1 DGSWP as an OT approximation

We begin by examining the illustrative scenario shown in Fig. 1, where the task is to compute an optimal transport (OT) plan between a mixture of eight Gaussians (source) and the Two Moons dataset (target). While some of the Gaussian modes are properly matched by min-SWGG, others are matched to the more distant moon, due to information loss along the direction orthogonal to the projection. To address this limitation, we apply DGSWP with a neural network parameterizing the projection function $\phi$, enabling more expressive projections. This improvement is reflected in the quantitative results: the transport cost associated to the DGSWP plan is notably closer to the squared Wasserstein distance than that of the min-SWGG method.

This transport plan above is obtained using the approach outlined in Sec. 4. To further evaluate the impact of the variance reduction technique introduced in Eq. (8), we now repeat the same experiment across 10 different random initializations of the neural network. The average learning curves for the variants with and without variance reduction are shown in Fig. 3. The results clearly indicate that using variance reduction improves both the final transport cost and the stability of the learning process. Based on this evidence, we adopt the variance reduction strategy in all subsequent experiments.

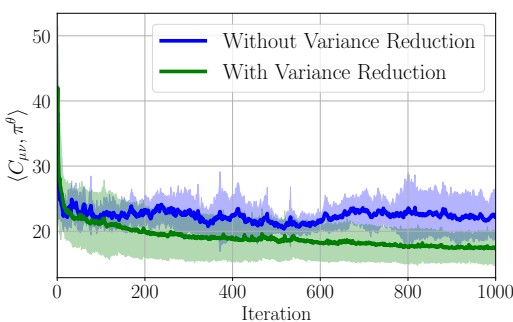

Figure 3: Impact of the variance reduction scheme (first 1,000 iterations).

## 5.2 Gradient flows

In these gradient flow experiments, our objective is to iteratively transport the particles of a source distribution toward a target distribution by progressively minimizing the DGSWP objective.

**Without manifold assumption.** We compare DGSWP with a linear and NN-based $\phi$ mapping against several baseline methods: Sliced Wasserstein distance (SWD), Augmented Sliced Wasserstein Distance, that has been shown in [12] to outperform GSW methods, and min-SWGG, evaluated in both its random search and optimization-based forms. Fig. 4 presents results across a range of target distributions designed to capture diverse structural and dimensional characteristics. In all experiments, the source distribution is initialized as a uniform distribution within a hypercube. We set the number of samples at $n = 50$, and repeat each experiment over 10 random seeds, reporting the median transport cost along with the first and third quartiles. We use the same learning rate for all experiments.

In two-dimensional settings, DGSWP consistently converges to a meaningful solution, whereas methods based on the min-SWGG objective sometimes fail to do so, even when using their original optimization scheme. More notably, in the high-dimensional regime, DGSWP stands out as the only slicing-based method capable of producing satisfactory transport plans, underscoring its robustness and scalability in challenging settings. This holds true even though min-SWGG is theoretically equivalent to Wasserstein when the dimension is high enough (here we have $d \geq 2n$); in practice, its optimization often struggles to find informative directions. In contrast, our proposed optimization strategy succeeds even in the linear case (where the formulation effectively reduces to min-SWGG) highlighting the practical benefit of our approach.

---

[3]`https://github.com/rtavenar/dgswp`

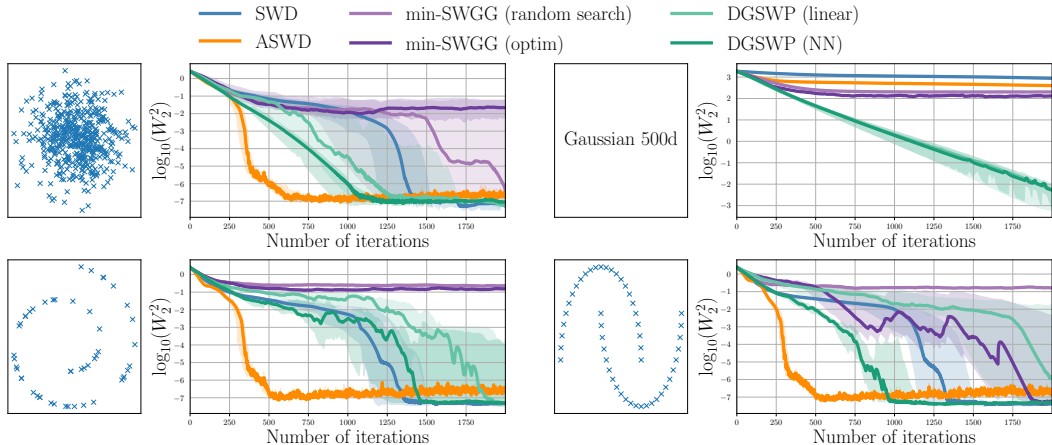

Figure 4: Log of the Wasserstein Distance as a function of the number of iterations of the gradient flow, considering several target distributions. The source distribution is uniform in all cases.

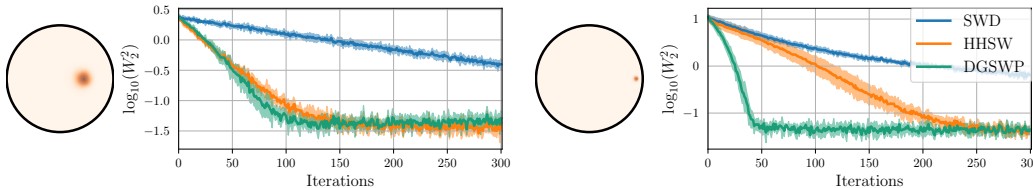

Figure 5: Log of the WD (second and fourth panels) for two different targets (first and third ones) as wrapped normal distributions for HHSW, SWD and DGWSP.

**On hyperbolic spaces.** Hyperbolic spaces are Riemannian manifolds of negative constant curvature [33]. They manifest in several representations and we consider here the Poincaré ball of dimension $d$, $\mathbb{B}^d$. We aim to construct a manifold feature map for hyperbolic space $\phi : \mathbb{B}^d \times \mathbb{B}^q \to \mathbb{R}$, typically relying on an analog of hyperplanes. Horospheres [26] generalize hyperplanes in the Poincaré ball and are parametrized by a base point located on the boundary of the space $\theta \in \partial \mathbb{B}^d = \mathbb{S}^{d-1}$. Akin to [28], we consider a map that corresponds to horosphere projection $\phi(x, \theta) = \log(\|x - \theta\|_2^2) - \log(1 - \|x\|_2^2)$.

Similary to [9], we assess the ability of generalized sliced Wasserstein plans to learn distributions that live in the Poincaré disk. We compare with the Horospherical sliced-Wasserstein discrepancy (HHSW, [9]) and Sliced Wasserstein computed on the Poincaré ball. For the gradient estimation, we rely on the von Mises-Fisher which is a generalization of a Gaussian distribution from $\mathbb{R}^d$ to $\mathbb{S}^{d-1}$ and perform a Riemannian Gradient Descent using Python toolbox `geoopt` [29] to optimize on $\theta$ and the source data (that live on $\mathbb{B}^d$). Figure 5 plots the evolution of the exact log 10-Wasserstein distance between the learned distribution and the target, using the geodesic distance as ground cost. We use wrapped normal distributions as source and set the same learning rate for all methods. DGWSP enables fast convergence towards the target, demonstrating its versatility for hyperbolic manifolds.

### 5.3 Sliced-OT based Conditional Flow Matching

We now investigate the use of DGSWP in the context of generative modeling. In the flow matching (FM) framework, a time-dependent velocity field $u_t(x)$ is learned such that it defines a continuous transformation from a prior to the data distribution. In practice, $u_t$ is trained by minimizing a regression loss on synthetic trajectories sampled from a known coupling between source and target distributions that determines the starting and ending points $(x_0, x_1)$ of the trajectory. Several variants of flow matching have been proposed depending on how these pairs are sampled. In Independent CFM (I-CFM, [34]), pairs are sampled independently from the prior and data distributions respectively. However, this approach ignores any explicit alignment between source and target samples. To address this, OT-CFM [54] proposes to deterministically couple source and target

samples using mini-batch OT, so that $(x_0, x_1) \sim \pi$ where $\pi$ is the OT plan computed between a batch of prior samples and a batch of data samples. This coupling tends to straighten the diffusion trajectories, which leads to improved generation quality in few-step sampling regimes.

Despite its advantages, OT-CFM relies on a trade-off: since computing exact OT is infeasible for large datasets, mini-batch OT is used, which leads to imperfect matchings, especially in high-dimensional spaces involving complex distributions, for which a mini-batch is unlikely to be representative of the whole distribution. This motivates the use of DGSWP as an alternative. By estimating transport plans using DGSWP, we can leverage significantly larger batches, resulting in better couplings, while maintaining low computational cost. In our experiments, we aggregate samples from 10 minibatches to compute the transport plan, and find that the additional computational cost remains negligible. Our approach is also supported by the findings of Cheng and Schwing [13], who show that increasing the batch size improves performance for OT-CFM in the few-sampling-steps regime.

| Integration method $\rightarrow$ | Euler | | DoPri5 | |
|---|---|---|---|---|
| Algorithm $\downarrow$ | FID | NFE | FID | NFE |
| I-CFM | 4.63 | 100 | 3.65 | 138.16 |
| OT-CFM | 4.82 | 100 | 3.86 | 132.90 |
| DGSWP-CFM (linear) | 4.17 | 100 | 4.47 | 110.37 |
| DGSWP-CFM (NN) | **3.56** | 100 | 3.87 | 120.04 |

Table 1: FID score and average number of function evaluations (NFE) per batch.

We conduct experiments on CIFAR-10 [31], reporting FID scores for DGSWP-CFM, OT-CFM, and I-CFM across varying numbers of sampling steps. We use the experimental setup and hyperparameters from Tong et al. [54]. For DGSWP, we evaluate both a linear projector and a more expressive non-linear embedding implemented by a neural network. We compare a fixed-step Euler solver with the adaptive Dormand-Prince method [19] for trajectory integration. The results in Table 1 underscore the importance of learning straight transport trajectories, which translates to more efficient generation: models that induce straighter flows require significantly fewer function evaluations (NFE) to achieve high-quality samples. Notably, even a simple 100-step Euler scheme proves highly effective when used with DGSWP-CFM, demonstrating the practicality of the method for fast generation.

A closer analysis of the results leads to three key observations: (i) Among the projection choices, linear DGSWP underperforms compared to its neural network-based counterpart, highlighting the benefit of extending min-SWGG with non-linear projections; (ii) When using the adaptive Dormand–Prince solver, DGSWP achieves performance comparable to OT-CFM in terms of FID, but with lower computational cost as indicated by the reduced number of function evaluations; (iii) In the fixed-step regime, DGSWP clearly outperforms all baselines under the Euler solver, offering the best trade-off between sample quality and efficiency, as illustrated in Fig. 6.

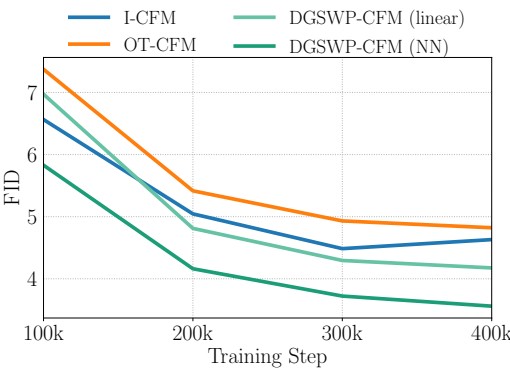

Figure 6: FID as a function of training iterations for various algorithms using 100-step Euler sampling.

## 6 Conclusion

This paper presents a novel differentiable approach to approximate sliced Wasserstein plans, incorporating non-linear projections. Its differentiable and GPU-efficient formulation enables the definition of optimal projections. The proposed method offers several key advantages: i) efficient computation comparable to that of SWD, ii) ability to provide an approximated transport plan, akin to min-SWGG, iii) improved performance in high-dimensional settings thanks to its improved optimization strategy iv) the capacity to handle data supported on manifolds through a generalized formulation. To the best of our knowledge, we also provide the first empirical evidence that slicing techniques can be effectively used in conditional flow matching, resulting in better performance and fewer function evaluations.

A key strength of slicing-based approaches for approximating OT lies in their favorable statistical properties, such as improved sample complexity [40]. These advantages also extend to projection-based methods on $k$-dimensional subspaces [42], as well as to min-SWGG [36]. Future works will explore

the conditions that the projection map $\phi$ must satisfy to preserve these statistical guarantees. Additionally, ensuring the injectivity of $\phi$ is essential for DGSWP to define a proper distance. Investigating injective neural architectures, such as those proposed in [46], is a promising research direction.

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

# A Appendix / supplemental material

## A.1 Proofs

We start by the proof of Lemma 1, concerned with 0-homogeneity of our target $\theta \mapsto h(\theta)$.

*Proof of Lemma 1.* We prove in fact that for all $c > 0$, $\pi^\theta = \pi^{c\theta}$. This is a consequence of the fact that if $\theta \mapsto \phi(x, \theta)$ is 1-homogeneous, then so does $\theta \mapsto C^\phi_{\mu\nu}(\theta)$. Thus, the level set of $\pi \mapsto g(\pi, c\theta)$ are the level-set of $\pi \mapsto g(\pi, \theta)$ dilated by a factor $c$. Hence, they share the same minimizers, and in consequence $h(c\theta) = h(\theta)$. $\qquad\square$

For the sake of completeness, we also prove the comment stating that a gradient flow on $h$ preserves the norm of the initialization with the following lemma

**Lemma 3.** *Let $U \subseteq \mathbb{R}^q$ an open set, assume that $h : U \mapsto \mathbb{R}$ is differentiable and $h$ is 0-homogeneous, i.e., $h(c\theta) = h(\theta)$. Consider the gradient flow dynamics*

$$\begin{cases} \theta(0) & = \theta_0 \in U \\ \dot{\theta}(t) & = -\nabla h(\theta(t)). \end{cases} \tag{9}$$

*Then, there exists an interval $I \subseteq \mathbb{R}_{\geq 0}$ and a unique solution $t \mapsto \theta(t)$ such that for all $t \in I$, $\langle \dot{\theta}(t), \theta(t) \rangle = 0$ and $\|\theta(t)\| = \theta_0$.*

*Proof. Orthogonal gradient of 0-homogeneous function.* Consider the real function $\psi : \mathbb{R} \to \mathbb{R}$ defined by $\psi(c) = h(c\theta)$. By 0-homogeneity, $\psi(c) = h(\theta) = \psi(1)$. Hence, $\psi$ is constant on $\mathbb{R}^*$, thus $\psi'(c) = 0$ for all $c \neq 0$. Using the chain rule for real function, we have that for all $c \neq 0$

$$\psi'(c) = \langle (c \mapsto c\theta)'(c), \nabla h(c\theta) \rangle = \langle \theta, \nabla h(c\theta) \rangle = 0.$$

Using this fact for $c = 1$, we conclude that

$$\forall \theta \in U, \quad \langle \theta, \nabla h(\theta) \rangle = 0. \tag{10}$$

*Orthogonal dynamics.* The existence and uniqueness of the Cauchy problem (9) comes from the Cauchy-Lipschitz theorem. Consider this solution $t \mapsto \theta(t)$ defined over $I$. Then,

$$\langle \dot{\theta}(t), \theta(t) \rangle = -\langle \nabla h(\theta(t)), \theta(t) \rangle = 0,$$

using (10). Hence, $\dot{\theta}(t) \perp \theta(t)$ for all $t \in I$.

*Conservation of the norm.* Consider $r(t) = \|\theta(t)\|^2$. The chain rule tells us that for all $t \in I$,

$$r'(t) = 2\langle \dot{\theta}(t), \theta(t) \rangle,$$

hence $r' = 0$ and thus $r$ is a constant function. $\qquad\square$

The proof of Proposition 1 relies on the fact that the $p$-Wasserstein distance is a metric on 1D measures, and that $\phi^\theta$ is conveniently supposed to be injective over the reference set $\mathcal{X}$.

*Proof of Proposition 1.* Let $p > 1$, $\theta \in \mathbb{R}^q$, and assume that $\phi^\theta$ is an injective map from $\mathbb{X}$ $\mathcal{X}$? to $\mathbb{R}$. Let $\mu, \nu, \xi$ three discrete distributions in $\mathcal{P}(\mathcal{X})$. Recall that

$$d^\theta(\mu, \nu) = W_p(\phi^\theta_\sharp \mu, \phi^\theta_\sharp \nu).$$

Using the fact that $W_p$ is a metric on $\mathcal{P}_p(\mathbb{R})$, we also obtain that $W_p$ is a metric on $\mathcal{P}(\mathcal{X})$ as a restriction.

**Well-posedness.** Since $d^\theta(\mu, \mu) = W_p(\phi^\theta_\sharp \mu, \phi^\theta_\sharp \mu)$, and that $W_p$ is a metric, we have that $d^\theta(\mu, \mu) = 0$.

**Symmetry.** The symmetry comes directly from the one of $W_p$.

**Positivity.** Suppose that $\mu \neq \nu$. Since $W_p$ is a metric, we only need to prove that $\phi^\theta_\sharp \mu \neq \phi^\theta_\sharp \nu$. Assuming that, where $x_i \neq x_{i'}$ for all $i \neq i'$ and $y_j \neq y_{j'}$ for all $j \neq j'$,

$$\mu = \sum_{i=1}^n a_i \delta_{x_i} \quad \text{and} \quad \nu = \sum_{j=1}^m b_j \delta_{y_j},$$

we have that

$$\phi^\theta_\sharp \mu = \sum_{i=1}^n a_i \delta_{\phi^\theta(x_i)} \quad \text{and} \quad \phi^\theta_\sharp \nu = \sum_{j=1}^m b_j \delta_{\phi^\theta(y_j)}.$$

Using the injectivity of $\phi^\theta$, we have that $\phi^\theta(x_i) \neq \phi^\theta(x_{i'})$ for all $i \neq i'$ and $\phi^\theta(y_j) \neq \phi^\theta(y_{j'})$ for all $j \neq j'$. Hence, $\phi^\theta_\sharp \mu = \phi^\theta_\sharp \nu$ if, and only if, $n = m$ and there exists a permutation $\sigma : \{1, \ldots, n\} \to \{1, \ldots, n\}$ such that

$$a_i = b_{\sigma(i)} \quad \text{and} \quad \phi^\theta(x_i) = \phi^\theta(y_{\sigma(i)}), \text{ for all } i.$$

But then, using the injectivity of $\phi^\theta$, we have $\{x_i\}_{i=1}^n = \{y_i\}_{i=1}^n$. Hence, $\mu = \nu$ which is a contradiction.

**Triangle inequality.** We have that $d^\theta(\mu, \nu) = W_p(\phi^\theta_\sharp \mu, \phi^\theta_\sharp \nu)$. Using the triangle inequality on $W_p$, we have that $d^\theta(\mu, \nu) \leq W_p(\phi^\theta_\sharp \mu, \phi^\theta_\sharp \xi) + W_p(\phi^\theta_\sharp \xi, \phi^\theta_\sharp \nu) = d^\theta(\mu, \xi) + d^\theta(\xi, \nu)$. $\qquad \square$

We now turn to the Stein's lemma. The proof of the Stein's lemma under a (weak) differentiability criterion [51] is classic, and relies on an integration by part and the properties of the normal distribution. Nevertheless, we are concerned with a function $\theta \mapsto h(\theta)$ that typically will have discontinuities, breaking the classical proof. Note that one cannot expect the Stein's lemma to hold true for any kind of discontinuities, even with almost everywhere differentiability. The celebrated example is the Heaviside function $h(\theta) = 1_{\theta \geq 0}$ in 1D where the Stein's lemma needs a correction term if there is a non-negligible number of them in sense of the $\mathcal{H}^{q-1}$ Hausdorff dimension. This setting was studied by [25, 53, 24] for various applications in statistics and signal processing, in particular for Stein's unbiased risk estimation.

*Proof of Lemma 2.* The proof of this result is mostly contained in [24, Proposition 1]. We outline the strategy. Assumption 1 requires to have $\mathcal{H}^{q-1}(\mathbb{R}^q \setminus C) = 0$. Hence, for all $i \in \{1, \ldots, q\}$ and Lebesgue almost all $(\theta_1, \ldots, \theta_{i-1}, \theta_{i+1}, \ldots, \theta_q) \in \mathbb{R}^{q-1}$, the map

$$t \mapsto h(\theta_1, \ldots, \theta_{i-1}, t, \theta_{i+1}, \ldots, \theta_q)$$

is absolutely continuous on every compact interval of $\mathbb{R}$. So, in turns, $h \in W^{1,1}_{\text{loc}}(\mathbb{R}^q)$ which in turns show that $h$ is almost differentiable in the sense of Stein [51] and we can thus apply [51, Lemma 1]. $\qquad \square$

We now turn to the proof of Proposition 2, regarding the properties of the smoothed version $h_\varepsilon$ of $h$.

*Proof of Proposition 2.* (*Admissibility.*) Let $\varepsilon > 0$ and $\theta \in \mathbb{R}^q$. Recall that

$$\pi^\theta_\varepsilon = \mathbb{E}_{Z \sim \mathcal{N}(0, I_q)} \left[ \arg \min_{\pi \in U(a,b)} g(\pi, \theta + \varepsilon Z) \right].$$

Denote by $u(z) = \arg \min_{\pi \in U(a,b)} g(\pi, \theta + \varepsilon z)$, hence $\pi^\theta_\varepsilon = \mathbb{E}_{Z \sim \mathcal{N}(0, I_q)}[u(Z)]$. For all $z \in \mathbb{R}^q$, $u(z) \in U(a,b)$ by definition of the minimization problem. Hence,

$$\pi^\theta_\varepsilon = \int_{\mathbb{R}^q} u(z) \rho(z) \mathrm{d}z,$$

where $\rho(z) = (2\pi)^{-q/2} \exp(-\frac{1}{2} \|x\|^2)$ is the probability density function of the multivariate normal law and $\mathrm{d}z$ is the Lebesgue measure on $\mathbb{R}^q$. Since $U(a,b)$ is convex and $u(z) \in U(a,b)$, then $\int_{\mathbb{R}^q} u(z) \rho(z) \mathrm{d}z \in U(a,b)$ also. In turn, since $\pi^\theta_\varepsilon \in U(a,b)$, then given the true solution $\pi^\star_{\text{OT}}$ of (1), we have $\sum_{i=1}^n \sum_{j=1}^m \pi^\star_{\text{OT},i,j} \|x_i - y_j\|^p_p \leq \sum_{i=1}^n \sum_{j=1}^m \pi^\theta_{i,j} \|x_i - y_j\|^p_p$

(*Differentiability.*) This is a direct consequence of Lemma 2 and Lebesgue dominated convergence theorem to invert expectation and derivative.

(*Consistency.*) The first fact is a consequence of Lebesgue dominated convergence theorem. The second one use the expression of the gradient through the variance reduction expression:

$$\nabla_\theta h_\varepsilon(\theta) = \varepsilon^{-1} \mathbb{E}_Z[(h(\theta + \varepsilon Z) - h(\theta))Z] = \mathbb{E}_Z\left[\frac{h(\theta + \varepsilon Z) - h(\theta)}{\varepsilon} Z\right].$$

Hence, again using Lebesgue dominated convergence theorem, we have

$$\lim_{\varepsilon \to 0} \nabla_\theta h_\varepsilon(\theta) = \lim_{\varepsilon \to 0} \mathbb{E}_Z\left[\frac{h(\theta + \varepsilon Z) - h(\theta)}{\varepsilon} Z\right] = \mathbb{E}_Z\left[\lim_{\varepsilon \to 0} \frac{h(\theta + \varepsilon Z) - h(\theta)}{\varepsilon} Z\right].$$

Recognizing the directional derivative of $h(\theta)$ if $h$ is differentiable at $\theta$, we get that

$$\lim_{\varepsilon \to 0} \nabla_\theta h_\varepsilon(\theta) = \mathbb{E}_Z[\langle \nabla h(\theta), Z\rangle Z] = \nabla h(\theta).$$

(*Distance.*) We assume here that $\phi^\theta$ is injective on $\mathcal{X}$. We split the proof for $h$ (**1.**) and $h_\varepsilon$ (**2.**).

**1.** Proof that $(\mu, \nu) \mapsto h(\theta)(\mu, \nu)$ is a distance over $\mathcal{P}(\mathcal{X})$. The *positivity* comes from the suboptimality of $\pi^\theta(\mu, \nu)$, that is $h(\theta)(\mu, \nu) \geq W_p^p(\mu, \nu) > 0$ if $\mu \neq \nu$ (as $W_p$ is a metric itself). The *symmetry* comes from the fact that $C_{\mu\nu}$ is symmetric and that $\pi^\theta(\mu, \nu) = (\pi^\theta(\nu, \mu))^\top$. Regarding the *well-posedness*, since $\phi^\theta$ is injective, then $W_p(\phi_\sharp^\theta \mu, \phi_\sharp^\theta \mu) = 0$ and $\pi^\theta(\mu, \mu)$ is the identity matrix. Hence,

$$h(\theta)(\mu, \mu) = \langle C_{\mu\mu}, \pi^\theta(\mu, \mu)\rangle = \sum_{i=1}^n \sum_{j=1}^n (C_{\mu\mu})_{ij} \pi_{ij}^\theta(\mu, \mu) = \sum_{i=1}^n (C_{\mu\mu})_{ii} = 0,$$

since for all $i$, $(C_{\mu\mu})_{ii} = \|x_i - x_i\|_p^p = 0$. Concerning the *triangle inequality*, let $\mu_1, \mu_2, \mu_3 \in \mathcal{P}(\mathcal{X})$. Let us denote

$$\pi^{12} = \pi^\theta(\mu_1, \mu_2) \in \mathbb{R}^{n_1 \times n_2}, \quad \pi^{13} = \pi^\theta(\mu_1, \mu_3) \in \mathbb{R}^{n_1 \times n_3}, \quad \pi^{23} = \pi^\theta(\mu_2, \mu_3) \in \mathbb{R}^{n_2 \times n_3}$$

Using the specific structure of the 1D optimal transport [50], there exists a tensor $\Pi \in \mathbb{R}^{n_1 \times n_2 \times n_3}$ of order 3 such that admits $\pi^{12}$, $\pi^{13}$ and $\pi^{23}$ as marginals, that is

$$\begin{array}{lll} \forall i, j, & \pi_{i,j}^{12} & = \sum_{k=1}^{n_3} \Pi_{i,j,k} \\ \forall i, k, & \pi_{i,k}^{13} & = \sum_{j=1}^{n_2} \Pi_{i,j,k} \\ \forall j, k, & \pi_{j,k}^{23} & = \sum_{i=1}^{n_1} \Pi_{i,j,k}. \end{array}$$

Since this structure provides us a "gluing lemma", we continue the proof similarly to the standard proof of the triangular inequality of the Wasserstein distance.

$$\begin{aligned} (h(\theta)(\mu_1, \mu_3))^{1/p} &= \left(\langle C_{\mu_1\mu_3}, \pi^{13}\rangle\right)^{1/p} \\ &= \left(\sum_{i=1}^{n_1} \sum_{k=1}^{n_3} \pi_{ik}^{13} \|x_i - z_k\|_p^p\right)^{1/p} \qquad \text{by definition} \\ &= \left(\sum_{i=1}^{n_1} \sum_{j=1}^{n_2} \sum_{k=1}^{n_3} \Pi_{ijk} \|x_i - z_k\|_p^p\right)^{1/p} \qquad \text{as glue.} \end{aligned}$$

Using that $\|x_i - z_k\|_p^p \leq \|x_i - y_j\|_p^p + \|y_j - z_k\|_p^p$, we get that

$$(h(\theta)(\mu_1, \mu_3))^{1/p} \leq \left(\sum_{i=1}^{n_1} \sum_{j=1}^{n_2} \sum_{k=1}^{n_3} \Pi_{ijk}(\|x_i - y_j\|_p^p + \|y_j - z_k\|_p^p)\right)^{1/p}.$$

Applying now the Minkowski inequality, we obtain that

$$(h(\theta)(\mu_1, \mu_3))^{1/p} \leq \left(\sum_{i=1}^{n_1} \sum_{j=1}^{n_2} \sum_{k=1}^{n_3} \Pi_{ijk} \|x_i - y_j\|_p^p\right)^{1/p} + \left(\sum_{i=1}^{n_1} \sum_{j=1}^{n_2} \sum_{k=1}^{n_3} \Pi_{ijk} \|y_j - z_k\|_p^p\right)^{1/p}.$$

Using the fact that $\Pi$ has marginals $\pi^{12}$ and $\pi^{23}$, we get that

$$(h(\theta)(\mu_1,\mu_3))^{1/p} \leq \left(\sum_{i=1}^{n_1}\sum_{j=1}^{n_2}\pi_{ij}^{12}\|x_i-y_j\|_p^p\right)^{1/p} + \left(\sum_{j=1}^{n_2}\sum_{k=1}^{n_3}\pi_{jk}^{23}\|y_j-z_k\|_p^p\right)^{1/p}.$$

Hence,

$$(h(\theta)(\mu_1,\mu_3))^{1/p} \leq (h(\theta)(\mu_1,\mu_2))^{1/p} + (h(\theta)(\mu_1,\mu_2))^{1/p}.$$

**2.** Proof that $(\mu,\nu) \mapsto h_\varepsilon(\theta)(\mu,\nu)$ is a distance over $\mathcal{P}(\mathcal{X})$. The *well-posedness* comes from the fact that $h_\varepsilon(\theta)(\mu,\mu) = \mathbb{E}_Z[h(\theta+\varepsilon Z)(\mu,\mu)] = \mathbb{E}_Z[0] = 0$. The symmetry is also a direct consequence of the *symmetry* of $h(\theta)(\mu,\nu)$ and the *positivity* comes from the fact that the expectation of a positive quantity is positive (understood almost surely on $\mathbb{R}^q$). For the *triangle inequality*, we use the linearity of the expectation: let $\mu_1,\mu_2,\mu_3 \in \mathcal{P}(\mathcal{X})$. Then, for all $\theta \in \mathbb{R}^q$, and for all $z \in \mathbb{R}^q$, using the fact that $h(\theta+\varepsilon z)$ is a distance

$$h(\theta+\varepsilon z)(\mu_1,\mu_3)^{1/p} \leq h(\theta+\varepsilon z)(\mu_1,\mu_2)^{1/p} + h(\theta+\varepsilon z)(\mu_2,\mu_3)^{1/p}.$$

Hence, taking the expectation and using linearity gives that

$$\mathbb{E}_Z[h(\theta+\varepsilon Z)(\mu_1,\mu_3)^{1/p}] \leq \mathbb{E}_Z[h(\theta+\varepsilon Z)(\mu_1,\mu_2)^{1/p}] + \mathbb{E}_Z[h(\theta+\varepsilon Z)(\mu_2,\mu_3)^{1/p}].$$

$\square$

### A.2 Algorithm

Algorithm 1 describes a gradient descent method to perform the minimization of $h_\varepsilon$ using the Monte-Carlo approximation from Eq. (8).

---

**Algorithm 1** Monte-Carlo gradient descent of $h_\varepsilon(\theta)$

---

**Require:** $\theta_0 \in \mathbb{R}^q$, step size policty $\eta_t$, smoothing parameter $\varepsilon > 0$, number of Monte Carlo samples $N$, number of iterations $T$
1: **for** $t = 0$ to $T-1$ **do**
2:     Sample i.i.d. perturbation vectors $z_1,\ldots,z_N \sim \mathcal{N}(0,\mathrm{Id}_q)$
3:     **for** $k = 1$ to $N$ **do**
4:         Solve OT problems to obtain $\pi^{\theta_t+\varepsilon z_k}$ and $\pi^{\theta_t}$         ▷ using 1D OT solver
5:         $g_k \leftarrow \langle C_{\mu\nu}, \pi^{\theta_t+\varepsilon z_k} - \pi^{\theta_t} \rangle$
6:     $\widehat{\nabla}h_{\varepsilon,N}(\theta_t) \leftarrow \frac{1}{\varepsilon N}\sum_{k=1}^N g_k z_k$         ▷ approximate gradient
7:     $\theta_{t+1} \leftarrow \theta_t - \eta_t \widehat{\nabla}h_{\varepsilon,N}(\theta_t)$         ▷ update parameter
8: **return** $\theta_T$

---

### A.3 Additional results and experiment details

All experiments except the Conditional Flow Matching (CFM) were run on a MacBook Pro M2 Max with 32 GB of RAM. On this machine, Fig. 1 took approximately 3 minutes per run (10,000 iterations), Fig. 3 about 6 minutes for 10 runs (with two models trained sequentially, 1,000 iterations), Fig. 4 required roughly 30 minutes, and Fig. 5 took around 10 minutes in total (all models considered, 10 repetitions). The CFM experiments were dispatched over a GPU cluster composed of GPU-A100 80G, GPU-A6000 48 Go, with a total runtime of 130h for training and inference of all presented models. We estimate that the total compute time over the course of the project—including experimentation, debugging, and hyperparameter tuning—is approximately two orders of magnitude larger than the reported runtimes for CPU-based experiments, and one order of magnitude larger for the GPU-based experiments.

#### A.3.1 Hyperparameter settings

We report here the hyperparameter configurations used across the main experiments. Figures 1 and 3 correspond to the same experiment—Fig. 3 highlights early training dynamics, while Fig. 1 depicts results at convergence. The projection network used is a 3-layer MLP with ReLU activations: (with

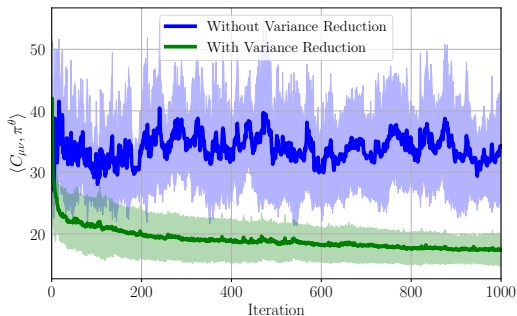

Figure 7: Impact of the variance reduction scheme from Eq. 8. Here, the same learning rate is used for both variants, in which case the variant without variance reduction does not even converge.

dimensions $2 \rightarrow 64 \rightarrow 16 \rightarrow 1$). Optimization is done using SGD with a learning rate of 0.2; for the variant without variance reduction, a lower learning rate of 0.0002 is used to ensure convergence (cf. Fig. 7 in which the same learning rate is used for both variants). In Figure 4 (gradient flow experiments), we perform 2000 outer flow steps using SGD with a learning rate of 0.01. At each flow step, we execute 20 projection steps (or inner optimization updates when using learnable projectors). For the latter, we use Adam with a learning rate of 0.01. The neural projector for our method is a single-hidden-layer MLP with ReLU activations and He initialization. In Figure 5, which investigates gradient flows on hyperbolic manifolds, we vary the outer learning rate across methods to account for differences in convergence speed: the base learning rate is 2.5, used for HHSW; SW uses a scaled learning rate of 17.5, and DGSWP uses a reduced rate of 0.83. Each flow step is composed of 100 projection or inner optimization steps. For the Conditional Flow Matching (CFM) experiment shown in Figures 6, 8, 9 and Table 1, we adopt the same training hyperparameters as in Tong et al. [54]. For our method specifically, the projection model is a 3-layer fully connected network with SELU activations: $3 \times 32 \times 32 \rightarrow 256 \rightarrow 256 \rightarrow 1$. Its parameters are optimized using Adam with a learning rate of 0.01. We perform 1000 optimization steps for the projection model at initialization, followed by 1 step per CFM training iteration.

### A.3.2 Additional results

Fig.8 presents a set of generated images using I-CFM, OT-CFM and DGSWP (NN) after 400k iterations using the 100-step Euler integrator.

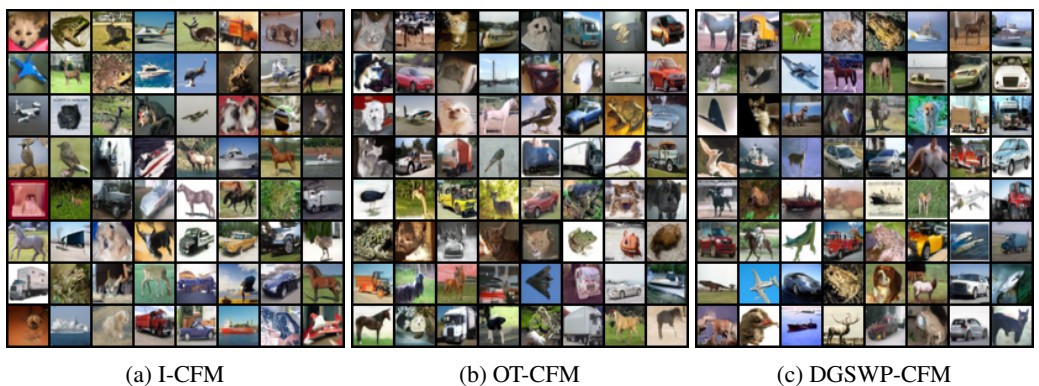

(a) I-CFM        (b) OT-CFM        (c) DGSWP-CFM

Figure 8: Example of generated images after 400k steps.

Fig. 9 presents the evolution of the image generation quality during training when using the adaptive-step Dormand-Prince strategy for the integration.

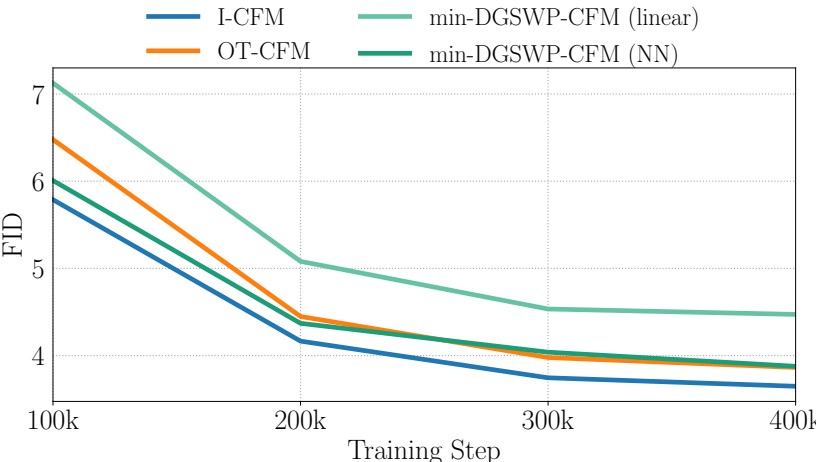

Figure 9: FID as a function of training iterations for various algorithms using adaptive-step DoPri5 sampling.

