# OpenReview forum: "Differentiable Generalized Sliced Wasserstein Plans"
_NeurIPS.cc/2025/Conference — NeurIPS 2025 poster_

### Official Review · Reviewer_enMR · 2025-06-26

**Clarity:** 3
**Significance:** 3
**Originality:** 3
**Rating:** 4
**Confidence:** 4

**Summary:**

This paper investigates the generalized sliced Wasserstein distance using arbitrary projection functions, including neural networks, thereby overcoming the limitations of previous methods that restricted the choice of projections. The core approach is based on a bilevel optimization framework combined with the application of Stein’s lemma for gradient estimation. Specifically, the authors formulate the minimization of the projection parameters θ as the upper-level problem, while the lower-level problem involves solving a sliced optimal transport (OT) problem with fixed θ. Due to the structure of sliced OT, the lower-level problem can be solved very efficiently using permutations. The gradient of the smoothed upper-level loss function is estimated via Stein’s lemma. To further accelerate convergence, the authors introduce a variance reduction technique leveraging the zero-mean property of Gaussian distributions. Numerical experiments on toy examples and flow matching demonstrate the improvements achieved by the proposed method.

**Questions:**

- What is the sample complexity for the gradient estimation of h, is this better than the previous work min-SWGG?

**Ethical Concerns:**

["NO or VERY MINOR ethics concerns only"]

**Limitations:**

The authors claim that a major drawback of min-SWGG is that the number of slices grows exponentially with the data dimension. However, the paper does not provide any analysis of the sample complexity for the proposed method, leaving it unclear whether the dimensionality issue has been addressed.
Although a variance reduction technique is proposed, the paper offers limited explanation or insight into its underlying mechanism.

**Paper Formatting Concerns:**

- line 198 "There exists a open set C" -> "There exists an open set C"

**Quality:**

3

**Strengths And Weaknesses:**

- The proposed bilevel formulation combined with the application of Stein’s lemma for gradient estimation is highly novel. Although bilevel optimization problems are typically challenging to solve, the authors leverage the closed-form solution of sliced OT, enabling efficient gradient estimation at the upper level.
- The introduced variance reduction technique substantially enhances computational efficiency.
- The numerical experiments provide convincing evidence of the method’s effectiveness.

---

> ### Author Rebuttal · Authors · 2025-07-29
>
> We would like to first thanks you for your praises of our work such as finding our bilevel formulation "_highly novel_" and our numerical experiments "_convincing evidence of the method’s effectiveness"_. We discuss below the main point that you addressed in your rebuttal regarding sample complexity.
>
> **Regarding sample complexity.** We are not entirely sure we fully understand your question. We provide a tentative answer addressing two different points, regarding two different sample complexities, and we hope this will offer enough clarity. We remain available during the discussion period to better develop one of these two points.
>
> - _Regarding the sample complexity of the Monte Carlo approximation of the gradient_.
> The Stein's approximation through DGSWP is done through a standard Monte Carlo approximation of the expectation. In this sense, the convergence rate of $\hat \nabla h_{\varepsilon, N}$ towards $\nabla h_\varepsilon$ has a rate of $O(1/\sqrt{N})$ where $N$ is the number of samples used (and crucially does _not_ depends on the dimension of the features). Note that more refined use of Monte Carlo methods, for instance by using MCMC methods could lead to a better rate such as $O(1/N)$. We leave this issue to future works.
>
> - _Regarding the sample complexity of (sliced-)Wasserstein and DGSWP_.
> The study of the statistical properties of (sliced-)Wasserstein has been the focus of considerable research in recent years. One significant result is that most sliced-based methods (including min-SWGG, see [1]) have a dimension-free sample complexity, which is not true for Wasserstein. This raises the question of the sample complexity of DGSWP. At this point, we do not have a definitive answer, as it depends on the function $\phi$ used to project the samples. If we consider a straight line, the dimension-free sample complexity of min-SWGG holds. Otherwise, it depends on the expressiveness of the neural network. For example, if the network can project samples onto a space of dimension $d \geq 2n$ where the samples are in general position, we would recover the sample complexity of Wasserstein. Then, we hypothesize that the resulting sample complexity will depend on the choice of the $\phi$ function. This is a very interesting question that we plan to explore in the near future.
>
> [1] Mahey, G. (2024). Unbalanced and linear optimal transport for reliable estimation of the Wasserstein distance (Doctoral dissertation, INSA Rouen Normandie).

---

### Official Review · Reviewer_8PTq · 2025-06-29

**Clarity:** 3
**Significance:** 3
**Originality:** 3
**Rating:** 5
**Confidence:** 3

**Summary:**

The authors reformulate Generalized Sliced Wasserstein Plans as a bilevel optimisation problem, generalising minimum Sliced Wasserstein Generalized Geodesics plans, to allow for non-linear slices parametrised by neural networks, in order to increase the expressiveness of their approximation to the true OT plan. The authors then propose a tractable optimisation scheme by Gaussian smoothing the outer objective of the bilevel problem, for which they provide a practical gradient estimator. They then validate their approach in variety of downstream applications.

**Questions:**

- I am not sure why there is a need to use (and to extend to the non-absolutely continuous) Stein’s lemma in section 4. To me, this appears to just be an application of REINFORCE/Evolutionary Strategies [1] which naturally handles the discontinuous setting. This citation should be added to direct readers to this literature, which could be used for improving the optimisation scheme proposed in the current paper.

- Are the authors able to provide any results/intuition for when DGSWP provides a tight approximation to the true OT distance/plan, akin to the results with min-SWGG? For instance, what performance can we expect to be able to achieve by scaling the neural network $\phi^\theta$?

[1] Salimans et al. (2017) Evolution Strategies as a Scalable Alternative to Reinforcement Learning

**Ethical Concerns:**

["NO or VERY MINOR ethics concerns only"]

**Final Justification:**

I believe the paper is worthy of acceptance and further the authors have address the points raised in the review. I will maintain my rating at 5 for acceptance.

**Limitations:**

The authors could have conduced more ablations for the stability of their optimisation scheme, for example, with respect to the hyper-parameters of $\epsilon$ and the number of Monte Carlo particles $N$.

**Quality:**

3

**Strengths And Weaknesses:**

Strengths:

- The paper is well written and does a good job at explaining the necessary background and their contribution.
- The proposed method is simple and appears easy to implement.
- The proposed method addresses an important problem in machine learning which is validated as an improved alternative to the commonly-used mini-batch OT in flow matching applications.

Weaknesses:

- The exact contribution of the authors could been signposted better - i.e. in section 3, the authors' reformulation in Definition 2 comes straight after the discussion of Generalized Sliced Wasserstein Plan which comes from previous work.
- It would have been nice to compare the performance of other slicing methods on the flow matching experiment. From looking at the previous experiments, it seems like DGSWP is especially strong in high-dimensional problems, so checking this on more realistic data would have been beneficial.
- In addition, for the flow matching experiments, it would be illuminating to report FID results over a range of NFE values for the Euler solver to demonstrate how each method improves the straightness of paths. This could also be achieved by looking at the path straightness metric from [1]

[1] Liu et al. (2022) Flow Straight and Fast: Learning to Generate and Transfer Data with Rectified Flow

Typos/Grammer:

Line 870: There is $\mathbb{X} \hspace{1mm} \mathcal{X}?$ which appears to be a comment that has been left in the appendix (I think the correct notation here should just be $\mathcal{X}$).

---

> ### Author Rebuttal · Authors · 2025-07-29
>
> We are grateful for the reviewer's very positive feedback and are pleased that our work was well received. We would like to take this opportunity to address your comments and improve the manuscript based on your thoughtful suggestions.
>
> **Regarding a better exposition of our contributions.** We will clarify our exact contribution more clearly. We want to emphasize that Definitions 1 and 2 are **new** objects that were never introduced in previous works. More precisely, [37] introduced SWGG that only tackles linear projection $\phi(x,\theta) = \langle x, \theta \rangle$, whereas [30] proposed _generalized sliced Wasserstein_ that is a distinct metric from the one proposed in our submission. We will rephrase the first paragraph of Section 3 to better clarify this point.
>
> **Regarding comparison with other slicing methods.** Except expected sliced Wasserstein (which we did not consider in the CFM experiment due to performance issues), sliced-based methods do not provide a transport plan. For CFM, getting such a plan is mandatory to sample trajectories. Then, they cannot be considered in this context.
> One major strength of DGSWP is its ability to provide good results in high dimensional problems, you are absolutely right. In the CFM experiments, the samples are set of images, one sample being one image seen as a vector of dimension 32 pixels $\times$ 32 pixels $\times$ 3 colors = 3072, which remains a challenging setting. Because we do not have access to extensive computation resources, we did not consider higher dimensional settings.
>
> **Regarding NFE values.** We run additional experiments for a range of NFE values, which confirms the good performance of DGSWP. The following table gives the average values of FID over 3 runs +- std:
>
> | Euler  (FID) for -> |     NFE = 20     |         NFE = 50 |        NFE = 100 |
> |:------------------- |:----------------:| ----------------:| ----------------:|
> | I-CFM               |   8.33 +- 0.09   |     5.49 +- 0.02 |     4.61 +- 0.06 |
> | OT-CFM              |   8.34 +- 0.07   |     5.78 +- 0.03 |     4.75 +- 0.06 |
> | DGSWP-CFM (linear)  |   8.57 +- 0.13   |     5.49 +- 0.01 |     4.16 +- 0.13 |
> | DGSWP-CFM (NN)      | **7.94 +- 0.17** | **4.72 +- 0.04** | **3.67 +- 0.03** |
>
> Regarding the ability of DGSWP to provide straight plans, it is indeed a desiderable behavior that is worth being investigating. Nevertheless, we believe that this question deserves a deeper investigation before raising complete conclusions, and we stick to the goal of this paper, which is to assess the ability of DGSWP to provide a meaningful plan.
>
> **Regarding evolutionary strategies.** One of the points of our submission is to show that a typical gradient descent-type algorithm can be used to compute DGSWP. We fully agree that different methods could be considered, for instance evolutionary strategies or reinforcement learning methods. We want to point out that [37] used a 0-order method to compute minSWGG (that boils down to GSWP with a linear projection). We will add a comment that mention the work you indicated as a potential alternative to design an algorithm to solve the bilevel problem behind GSWP.
>
> **Regarding scaling.** This is indeed an interesting question, thank you. It has been shown that, provided that i) we are able to sample the direction $\theta$ that minimizes SWGG ii) the samples are in general position (see [2]), min-SWGG is equal to Wasserstein when the dimension $d\geq 2n$. As DGSWP allows optimizing over $\theta$ (or a non-linear version of it), it is therefore natural to wonder under which conditions one could recover this result. However, it is hard to give a definite answer to it, for several reasons:
> - as we face a non-convex problem, it is hard to guarantee that we will converge to the global optima
> - to our knowledge, there is no network that guarantees that the embeddings will be in general position
>
> Nevertheless, one can rely on this result to design a neural network architecture that is likely to provide good results, that is to say expressive enough to provide an embedding space in a large enough dimension. We propose to add a discussion about this on the paper.
>
>
> [2] Cover, T. M. (1967). The number of linearly inducible orderings of points in d-space. SIAM Journal on Applied Mathematics, 15(2), 434-439.

---

> > ### Comment · Reviewer_8PTq · 2025-08-05
> >
> > Thank you for the response to the points raised in the review. This has answered all the question I have asked. I will maintain my recommendation of acceptance.

---

### Official Review · Reviewer_WETw · 2025-07-02

**Clarity:** 3
**Significance:** 2
**Originality:** 2
**Rating:** 4
**Confidence:** 4

**Summary:**

The paper introduces a differentiable approximation of OT distance and plan using generalized Sliced-Wasserstein projections and Sliced-Wasserstein Generalized Geodesics framework.

**Questions:**

Regarding the weaknesses raised, I request clarification on the following points:

1. Injective Map: Could the authors specify the construction of the injective map $\phi^\theta$ used in the experiments? Additionally, please indicate where this component is implemented in the code supplement. This would help verify whether the assumption required by Proposition 2.4 is satisfied in practice.

2. Choice of Conditional Flow Matching (CFM): While we agree that novel methods need not outperform the state of the art, the rationale for selecting Conditional Flow Matching as the sole generative modeling framework is currently underexplained. Could the authors elaborate on the motivation and intuition behind this design choice? A clearer justification would help contextualize its relevance and potential benefits.

3. Runtime Comparison with Standard OT Solvers: Since a central objective of the paper is to approximate both the OT distance and its corresponding transport plan, please provide a runtime and memory usage comparison between the proposed method and a standard OT solver (e.g., the network simplex or Sinkhorn algorithm). These two factors are critical for evaluating the method’s practical efficiency. Such a comparison—conducted under controlled settings with varying numbers of support points and data dimensionality—would provide valuable insight into the computational trade-offs involved in adopting the proposed approach over classical solvers.

4. CIFAR-10 Evaluation Details: For the CIFAR-10 generative modeling task, please clarify (i) the number of independent runs used to compute the reported performance metrics, and (ii) the runtime required for the baseline methods used in comparison. This information is important for assessing the statistical reliability and computational efficiency of the proposed method.

---

I would be willing to consider increasing my overall evaluation if the authors are able to clarify the points raised above.

**Ethical Concerns:**

["NO or VERY MINOR ethics concerns only"]

**Final Justification:**

I thank the authors for their response. I believe all of my concerns have been adequately addressed. I am now satisfied with the clarifications and revisions provided, and I have updated my score in favor of acceptance.

**Limitations:**

.

**Quality:**

3

**Strengths And Weaknesses:**

**Strengths.**

- The paper is largely self-contained. The first two sections provide a clear and comprehensive introduction to related work on Optimal Transport (OT), Sliced OT (or Sliced Wasserstein, SW), and its extensions—namely Generalized SW (GSW) [1] and SW Generalized Geodesics (min-SWGG) [2]. These prior formulations are thoroughly recalled and serve as a strong foundation for the developments in later sections.

- The proposed method in Sections 3 and 4 appears to be novel to the best of my knowledge. It is accompanied by theoretical guarantees, with detailed proofs provided in Appendix A.1. The overall methodology is clearly described, including a step-by-step algorithm in Appendix A.2, which I believe is sufficient for readers to understand and implement the proposed framework.

- The experimental results presented in Section 5 support the claims made, and the authors provide code in the supplementary material (though I have not verified it).

**Weaknesses.**

- The method proposed in Section 3 appears to be a straightforward combination of two prior works: Generalized Sliced Wasserstein (GSW) for projection, and min-SWGG for selecting the transport plan with minimal cost. Proposition 1 and Lemma 1 can be viewed as direct consequences of these two existing methods.

- Section 4 introduces a differentiable formulation for transport plan estimation. However, the construction of the injective projection map $\phi^\theta$, which is essential to both the theoretical results (e.g., Proposition 2.4) and the experiments, is not explicitly defined. While injective maps can be constructed (e.g., via $f:x\mapsto(x,f(x))$), the paper does not provide details or evidence that the maps used in experiments satisfy the injectivity conditions required for the proposed distance to be valid.

- Regarding the experimental evaluation: I appreciate the first experiment for its clarity. The second experiment follows a common setup seen in many related works and, while somewhat standard, still adds value. However, my concerns lie mainly with the third experiment involving generative modeling on CIFAR-10. Several recent works in OT and SW-based generative modeling have achieved substantially better FID scores on this dataset using alternative frameworks (e.g., [3]). The authors’ decision to constrain their method to Conditional Flow Matching is not well-justified.

- Additionally, while the paper reports the total runtime for the proposed method, it does not provide any runtime comparisons with the baselines. For reference, the CIFAR-10 experiment reportedly takes over 130 hours (>5 days) on a cluster of A100s and A6000s—though the exact number of GPUs is not disclosed—which raises concerns about efficiency. Lastly, the paper does not report standard deviations or the number of runs used to compute the reported results in this task, limiting reproducibility and statistical confidence.

---
**Reference.**

[1] Soheil Kolouri et al., Generalized Sliced Wasserstein Distances

[2] G Mahey et al., Fast optimal transport through sliced generalized wasserstein geodesics

[3] Khai Nguyen et al., Sliced Wasserstein with Random-Path Projecting Directions

---

> ### Author Rebuttal · Authors · 2025-07-29
>
> We thank the reviewer for their comments. We remain available throughout the discussion period for any further clarifications, and we hope you will consider increasing your overall evaluation in the light of these clarifications.
>
> **Regarding the difficulty of our contribution.** We thank the reviewer for highlighting the connection with GSW  [37] and minSWGG  [30].  While our approach builds on these ideas, and we obviously do not deny it, it is not a simple juxtaposition. The key contribution is to view this generalization of minSWGG as a bilevel optimization problem, and to carefully design a gradient-like method to solve it. It is well known that bilevel optimization in the context of non-strongly convex inner problem is difficult (see e.g., [1]) and we believe that our approximation results of GSWP are non-trivial. We agree that the proofs of Lemma 1 and Proposition 1 do not present technical difficulties, but we believe it is still important to formally state them, as they are necessary first steps before stating results on DGSPW. Despite these simple proofs, the approximation results that follow in Section 4 are less classical, and we believe that the proof of Proposition 2 contains interesting arguments (especially the fact that we obtain a gluing lemma "for free").
>
> [1] Bolte, J., Lê, Q. T., Pauwels, E., & Vaiter, S. (2025). Geometric and computational hardness of bilevel programming. Mathematical Programming, 1-36.
>
> **Regarding the injectivity condition.** We would like to recall that there is no injective map from $\mathbb{R}^d \to \mathbb{R}$ for $d > 1$, and we emphasize that we require only injectivity from the (potential) support to $\mathbb{R}$. This condition is satisfied almost surely in the linear case (all points in general configuration), and could be also satisfied with injective neural network [2]. Note that we require this property to prove that we obtain a metric, the object $d^\theta$ _still_ make sense without this hypothesis. In particular, we did not observe performance collapse due to noninjective $\phi$ map, and we believe that it due that it is unlikely that the training procedure select a MLP that is degenerate with regard to this property. We do not know however to prove such result, since it involves structural properties of the training dynamics of neural networks. From a numerical point of view, we simply rely on the order given by the sort algorithm used by Python that can be interpreted as a selection map from the set of minimizers.
>
> [2] Puthawala, Michael, et al. "Globally injective relu networks." _Journal of Machine Learning Research_ 23.105 (2022): 1-55.
>
> **Regarding the choice of Conditional Flow Matching.** The goal of the experimental section is to assess the performance of DGSWP in various contexts. The first experiment aims to provide a qualitative evaluation of the plan, while the second examines the convergence properties of DGSWP. Lastly, we seek to evaluate our method in a setting where a transport plan is required, and the number of samples is large. In this context, mini-batch OT is the main competitor, as, except for the expected sliced Wasserstein (see General comment), no sliced methods provide a transport plan. For these reasons, and to challenge DGSWP in a context where slicing methods have not been considered before, we focus on CFM, as [3] is well-positioned in the literature. Therefore, the primary motivation was not to deliver state-of-the-art performance on a generative modeling task but rather to investigate the performance of DGSWP across different settings. We propose to clarify this in the final version of the paper.
>
> [3] Tong, A., Fatras, K., Malkin, N., Huguet, G., Zhang, Y., Rector-Brooks, J., ... & Bengio, Y. (2024). Improving and generalizing flow-based generative models with minibatch optimal transport. Transactions on Machine Learning Research, 1-34.
>
> **Regarding runtime comparison with OT solvers.** You are perfectly right, we have not explicitly stated the complexity for our DGSWP method in the paper. We will fix that in the final version of the manuscript upon acceptance.
> Let $n$ be the number of samples in each distribution, $d$ be the dimensionality of the input space, $m$ be the number of Monte Carlo samples used for the gradient estimation in Eq. (8) and $p$ the number of gradient steps required to reach convergence, then the time complexity for the linear DGSWP is $O(p m n \log n + p m n d))$.
> Upon acceptance of the manuscript, we will add a visualization of the corresponding running times for both linear and NN variants of DGSWP.
>
> **Regarding CIFAR-10 evaluation.** Results were originally reported for one single run; we performed additional experiments to report results computed on 3 runs:
>
> |                    |   Euler  (FID)   |           DoPri5 FID (NFE) |
> |:------------------ |:----------------:| --------------------------:|
> | I-CFM              |   4.61 +- 0.06   |  3.74 +- 0.04 (6896 +- 47) |
> | OT-CFM             |   4.75 +-0.06    |  3.81 +- 0.04 (6490 +- 36) |
> | DGSWP-CFM (linear) |   4.16 +- 0.13   | 4.45 +- 0.06 (5556 +- 108) |
> | DGSWP-CFM (NN)     | **3.67 +- 0.03** |  3.95 +- 0.06 (5696 +- 64) |
>
> To achieve these results, we performed the computations on a cluster consisting of multiple GPUs (utilizing only 2 GPUs for each experiment) with different GPU types, using a job scheduler based on resource reservation and availability. As a result, it is challenging to provide a definitive answer to the question about the required runtime. However, we can assert that the CFM experiment with mini-batch OT and DGSWP has roughly the same runtime (as mentioned in the paper: "aggregate samples from 10 minibatches to compute the transport plan, and find that the additional computational cost remains negligible"). As for I-CFM, the running time is approximately 25% lower than that of OT-CFM.

---

> > ### Comment · Reviewer_WETw · 2025-08-03
> >
> > I thank the authors once again for their efforts in preparing the rebuttal. I understand that, due to the policy of the venue, some of the remaining concerns will be addressed upon acceptance. With the exception of the responses regarding the injectivity of the map $\phi^\theta$ and the runtime comparison, I am generally satisfied with the rest of the rebuttal.
> >
> > ---
> >
> > **Injectivity:** I would like to make a small correction: there do exist injective maps from $\mathbb{R}^d$ to $\mathbb{R}$ though they are not continuous. This is a well-known mathematical construction and does not affect the validity of the authors’ response.
> >
> > That said, I would like to follow up on a conceptual point. Since $\mathcal{X}$ is the set of all potential supports, why can it not be a manifold of dimension greater than one? This observation leads me to question the generality of the proposed approach. I do acknowledge that the definition of the proposed metric does not strictly rely on this injectivity assumption. However, based on the limited scope of the experimental section, I am inclined to interpret the paper as being more theoretical in nature. As such, clarifying this point would help strengthen the conceptual foundation of the work.
> >
> > **Runtime:** To be frank, I am not entirely satisfied with the authors' inability to provide runtime reports, especially considering that the experiments are conducted on CIFAR-10. While I understand the constraints on resources, CIFAR-10 is a standard benchmark whose scale is generally considered manageable even within the context of Optimal Transport × ML studies. Even if precise timing measurements are not available, I would appreciate at least a coarse estimation or qualitative comparison regarding how the proposed method performs in terms of speed relative to the baselines.
> >
> > ---
> >
> > Once again, I thank the authors for their rebuttal. Several points remain open, and I remain open to further discussion with the reviewers.

---

> > > ### Author Response · Authors · 2025-08-04
> > > **Post-rebuttal**
> > >
> > > We thank the reviewer for their constructive feedback.
> > >
> > > Regarding injectivity, we agree with the reviewer that an injective map from $\\mathbb{R}^d$ to $\\mathbb{R}$ exists. Our point in the rebuttal was more practical, ie. that no such injective map can be implemented by a standard neural network. We will clarify this in the final version.
> > >
> > > Concerning the runtime comparison on the CIFAR-10 CFM experiment, we can provide the following qualitative assessment: I-CFM is approximately 25% faster than the other methods. Meanwhile, OT-CFM, min-DGSWP-CFM (linear), and min-DGSWP-CFM (NN) exhibit comparable runtimes. In practice, the DGSWP batch size serves as a hyperparameter of our method, and we set it to 10 specifically to align the runtime of our approach with that of OT-CFM.

---

> > > > ### Comment · Reviewer_WETw · 2025-08-05
> > > >
> > > > I thank the authors for their response. I believe all of my concerns have been adequately addressed. I am now satisfied with the clarifications and revisions provided, and I have updated my score in favor of acceptance.

---

### Official Review · Reviewer_Q6uB · 2025-07-05

**Clarity:** 3
**Significance:** 3
**Originality:** 2
**Rating:** 4
**Confidence:** 4

**Summary:**

This paper introduces a new algorithm, Differentiable Generalized Sliced Wasserstein Plans (DGSWP), an extension of sliced-based approximations of Optimal Transport (OT). The conventional computation of OT is costly. Slicing algorithms like min-SWGG approximation address the problem by using one-dimensional optimal transport plan, which can be computed efficiently, as an approximation. However, current variants suffer from drawbacks such as linear projection dependence and inability to scale to large dimensions. The paper reframe min-SWGG into a bilevel optimization problem and present a differentiable approximation scheme using Gaussian smoothing and Stein's lemma for gradient-based optimization. The generalized formulation also supports non-linear projections (e.g., neural networks) and can be applied to manifold-valued data, like hyperbolic spaces. They empirically validate the method on three environments: non-linear toy problems for OT approximation, gradient flows in high-dimensional Euclidean and hyperbolic spaces, and a novel application to conditional flow matching for image synthesis, showcasing better sample efficiency and quality.

**Questions:**

1. Since the authors use control variate for Monte Carlo estimation of gradient, I wonder if control variates techniques in sliced Wasserstein cases be applicable here for the linear case?  [2] [3] It is just a minor aspect but it is worth to discuss.

[3] Sliced Wasserstein Estimation with Control Variates, Nguyen et al
[4] Sliced-Wasserstein Estimation with Spherical Harmonics as Control Variates, Leluc et al

2. Can the author verify the accuracy of the obtained plans with the true OT in some cases where the ground metric is known (Eculidean, manifold, non-manifold)? It is worth to add the comparison with expected sliced optimal transport plan. I will definitely raise my score if this question is addressed.

**Ethical Concerns:**

["NO or VERY MINOR ethics concerns only"]

**Final Justification:**

The authors addressed all my questions. I believe the paper addresses an important question in the literature of sliced optimal transport.

**Limitations:**

Yes

**Quality:**

3

**Strengths And Weaknesses:**

# Strengths

The paper  addresses an important question of obtaining transportation plans from sliced optimal transport, which is a valuable contribution to the field of computational optimal transport. Differentiable Generalized Sliced Wasserstein Plans (DGSWP) is  a method enabling efficient, differentiable OT plan approximation through a bilevel optimization reformulation of the min-SWGG problem. The paper addresses several major limitations of existing sliced OT methods, most notably linear projections and the lack of differentiability. By taking advantage of Gaussian smoothing and Stein's lemma, the authors construct a differentiable surrogate objective that can be optimized via stochastic gradient descent, even in high dimensions. The generalization to non-linear projections via neural networks and to manifold-valued data (e.g., hyperbolic spaces via horospherical slicing) demonstrates excellent flexibility. Experiments are well thought out and range across a range of settings, from toy transport problems, high-dimensional gradient flows, to a novel application of OT plans in conditional flow matching for generative models (which is new itsself). The empirical results improve over previous sliced OT methods across the board, especially in structured and high-dimensional settings.

# Weaknesses

While the proposed smoothed optimization method with Stein's lemma is elegant, the outer optimization problem remains non-convex and initialization-sensitive. An issue particularly in high-dimensional regimes where it is difficult to find informative directions. Experimentally, although the method shows empirical gains on conditional flow matching on CIFAR-10, the results are for single runs and without error bars or statistical significance tests, so it is hard to assess robustness. In addition, the paper misses comparison with a recent competitor,  the expected sliced optimal transport plan [1, 2], which averages sliced optimal transport plans with respect to a slicing distribution over projection parameter (which can be a direction or a neural network) instead of finding the minimum projection parameter. In addition, comparisons against more established differentiable OT solvers such as entropic Sinkhorn-based methods would provide a clearer picture of the scalability, accuracy, and plan quality trade-offs. Finally, while the theoretical formulation is comprehensive, some parts, especially the connection between smoothing methods and their variance, would benefit from more empirical exploration or ablation studies.

[1] Expected Sliced Transport Plans, Liu et al
[2] Energy-Based Sliced Wasserstein Distance, Nguyen et al

---

> ### Author Rebuttal · Authors · 2025-07-29
>
> We would like first to thanks the reviewer for their positive comments, acknowledging the quality of the writing and the novelty. We would like to address your main comments below.
>
> **Regarding statistical significance.** We have provided gradient flows results with several repetitions. For CFM, the paper contains results only for one run. We performed additional runs to better assess the variability of the results. Below, we report the average +- std for 3 runs, which keeps the conclusion unchanged.
>
> | |  Euler  (FID) | DoPri5 FID (NFE) |
> |:--------------- |:--------:| ---------------:|
> | I-CFM | 4.61 +- 0.06 | 3.74 +- 0.04 (6896 +- 47) |
> | OT-CFM | 4.75 +-0.06     | 3.81 +- 0.04 (6490 +- 36) |
> | DGSWP-CFM (linear) |  4.16 +- 0.13  | 4.45 +- 0.06 (5556 +- 108) |
> | DGSWP-CFM (NN)               |   **3.67 +- 0.03**      |   3.95 +- 0.06 (5696 +- 64)              |
>
> **Regarding Expected Sliced Optimal Transport.** When considering ESW as an additional competitor, we get the results from Tables S.1 and S.2 below:
> For the dataset in Fig. 1, varying the temperature parameter $\tau$ for Expected Sliced Wasserstein, we get the following results in terms of transportation cost:
> _Table S.1_
> | Method | $\langle \pi, C\rangle$ cost |
> |----|----|
> | Exact OT | 13.50 |
> | min-SWGG | 22.22 |
> | Expected Sliced Wasserstein ($\tau=0$) | 32.28 |
> | Expected Sliced Wasserstein ($\tau=1$) | 21.30 |
> | Expected Sliced Wasserstein ($\tau=10$) | 20.09 |
> | DGSWP (NN) | 13.80 |
>
> For the gradient flow experiment in Fig. 4, we get the following performance after 2,000 iterations for the Swiss Roll dataset (note that results are consistent across all 4 datasets).
>
> _Table S.2_
> | Method | $\log_{10} W$ (at convergence) |
> |---|---|
> | min-SWGG (optim) | -0.79 |
> | ESW ($\tau=0$) | -0.07 |
> | ESW ($\tau=1$) | -0.08 |
> | ESW ($\tau=10$) | -0.13 |
> | DGSWP (NN) | **-5.83** |
>
> In other words, in the experimental setup of Fig. 1 and Fig. 4, ESW is consistently outperformed by both min-SWGG and DGSWP in terms of transportation cost and convergence of the corresponding gradient flows (varying the hyper-parameter from $\tau=0$ to $\tau=10$).
> Based on these initial comparisons on toy datasets, we have decided not to include ESW in the comparison for larger-scale problems such as the CFM experiment.
>
> **Regarding control variate for Monte Carlo estimation of gradient.** Thanks for this comment. We believe that the line of works that you mention is very interesting. Overall, we kept a discussion on the variance reduction aspect quite minimalistic, both from a theoretical and practical aspects. We believe that finding better, or even optimal, variance reduction method working alongside with Stein's approximation is an interesting topic of future works.
>
> **Regarding assessing the quality of a transportation plan.** Assessing the quality of a transportation plan is hard. In this work, we do so by computing the cost associated to a plan, ie. $\langle \pi, C\rangle$ (cf. Fig 1). When doing so for the dataset in Fig. 1, varying the temperature parameter $\tau$ for Expected Sliced Wasserstein, we get the  results obtained above in Table S.2:
> Notably, we observe that the cost for ESW is consistently larger than that for min-SWGG (which is expected, since ESW averages plans over drawn directions while min-SWGG retains the direction of minimal cost), and in practice larger than that of DGSWP (NN).
> Note that one could argue that assessing the similarity of a transport plan with respect to the ground truth one could be done by computing the matrix norm of the differences between plans, but we observe in practice that such a distance tends to favor uninformative uniform plans over more meaningful sharp ones, which is why we avoid such metrics in our work.
> If the reviewer has suggestions for an alternative way to assess the quality of an estimated transport plan, we would be happy to provide such metrics for the compared methods during the author-reviewer discussion period.

---

> > ### Comment · Reviewer_Q6uB · 2025-08-02
> >
> > I would like to thank the authors for the response. Although there is still some missing aspects  in the paper like assessing the quality of a transportation plan, I will keep my positive score since I believe the paper pushes forward an important area of sliced optimal transport.

---

> > > ### Author Response · Authors · 2025-08-04
> > > **Post-rebuttal**
> > >
> > > We thank the reviewer for their positive evaluation and support of our work.
> > >
> > > We remain interested in their perspective on how to assess the quality of a transportation plan beyond our current cost-based strategy. In particular, we would appreciate any suggestions, especially given that naive comparisons between transportation plans (e.g., via matrix distances) can be misleading and may not reflect meaningful alignment.
> > >
> > > As an illustration for this claim, let us consider the following example.
> > > Let $\\mu_1$ and $\\mu_2$ be two discrete 1d distributions $\\mu_1 = [0, 1, 100]$ and $\\mu_2 = [.4, .6, 99]$ with uniform weights.
> > > The optimal transportation plan $\\pi^\\star$ in this setting is the scaled identity matrix.
> > > Now let us consider two alternative transportation plans: $\\pi_0$ is a uniform (hence totally uninformative) plan and $\\pi_1$ is:
> > > $$
> > >   \\begin{bmatrix}
> > >   0 & \\frac{1}{3} & 0 \\\\
> > >   \\frac{1}{3} & 0 & 0 \\\\
> > >   0 & 0 & \\frac{1}{3}
> > >   \\end{bmatrix}
> > > $$
> > > $\\pi_1$ successfully maps third atoms of the distributions together but it swaps mappings of first two atoms.
> > > In terms of transportation cost, $\\pi_1$ is much better than $\\pi_0$: $\\langle \\pi_0, C(\\mu_1, \\mu_2) \\rangle \\approx 4356.4$, $\\langle \\pi_1, C(\\mu_1, \\mu_2) \\rangle \\approx 0.6$. However, in terms of matrix norm, the opposite conclusion is reached: $\\pi_0$ is closer to $\\pi^\\star$ than $\\pi_1$.
> > > We hope this example clarifies why we believe that computing distances to the optimal transportation plan is not a reliable strategy for assessing plan quality and remain open to any alternative strategy.

---

### Official Review · Reviewer_kzcn · 2025-07-09

**Clarity:** 3
**Significance:** 3
**Originality:** 3
**Rating:** 5
**Confidence:** 4

**Summary:**

The paper is an extension of the min-SWGG
towards two direction: 1) generalization of the linear projections of the measure supports to more general function $\phi$, in particular neural networks, and 2) smoothing of the objective wrt $\pi$ based on (a generalization of) Stein's lemma. Finally DGSWP comes out.
Numerical examples on the approximation of the OT plans themselves (also on hyperbolic spaces)
and on their use in flow matching (also on hyperbolic spaces) are provided.

**Questions:**

Questions:
- Proposition 1 seems of limited practical use, as it requires an injective function from $\mathbb R^d$
to $\mathbb R$, which is numerically super-unstable.
- When $\phi$ is a NN with a ReLU activation function as in the experiments (see also Lemma 1 - smoother activation functions are rarely positive one-homogeneous), it is not differentiable
at hyperplanes in $\R^q$. Therefore Assumption 1 is likely not fulfilled.
- It is not clear if (linear) DGSW has practical benefits to SWGG.
Fig. 1 does not show the former, and Fig. 4 gives the error depending on the iteration number.
It would be better to have the computation time, as one interaction of DGSW needs the additional sampling of the Gaussian. In Fig. 4, ASWD appears without explanation, and seems to be the best algorithm in most cases.
- A comparison with expected plans from [35] might be interesting.
- Which function evaluations does Table 1 mean? There (and in Fig. 6), computation time would be more relevant than number of iterations/function evaluations.
- What about local minima in the minimization with respect to theta; there should my many high dimensions?
- Could also e.g. $d$-spheres be considered or is the positive curvature a problem?
(since I stepped in just as a emergency reviewer I had not enough time to go through the whole supplement)

I have some minor items:
- Until definition 2 you deal with $p>1$, why you move to $p \ge 0$ (and waht sense make $p=0$ at all),
moreover the setting for $p=\infty$ should be explained or excluded.
- line 121: ''extend the latter framework to by ...'' something is missing here
- line 132: ''The authors ...'' in the context this refers to 836] which is a PhD by one person
- line 151: why $g$ is continuous without any assumptions on $\phi$?
- line 152: waht is a suboptimal point?
- The formulation of Lemma 2 is not nice. The definition of $Z$ comes after its use
and one should write ''for any $\varepsilon >0$.
- caption Fig 2: ''hence there is the need ...''
- Lemma 3 (supplement): make comment in the main paper better visible (e.g. as a remark with a number to refer to or just as lemma)
- I do not like the notation $C_{\nu,\mu}$ since the cost function does not depend on the measures, but only
on their support, i.e. the $x_i$, $y_j$
- Please start personal names with capitals (Radon, Wasserstein, Sinkhorn, ... ) in the text and the references; same with names of journals

**Ethical Concerns:**

["NO or VERY MINOR ethics concerns only"]

**Final Justification:**

authors have addressed my concerns in a sufficient way, I raise the scaore to 5

**Limitations:**

see weakness and questions

**Quality:**

3

**Strengths And Weaknesses:**

Strengths:
It is a solid paper on a generalization of min-SWGG (min-SWGG does not work so well in our experience). The smoothing idea based on Stein's lemma is original and the performance appears to improve previous methods.

Weaknesses:
The writing is sometimes suboptimal, e.g. the key formula is (5). Here it would be good to see the definition of $g$ again or a link to formula (4), see also minor items below.
In the numerical part, I am missing some experiment that compares the computed approximate planes to the original ones and on the reduction of computation time achieved by the new method, see also below.

---

> ### Author Rebuttal · Authors · 2025-07-29
>
> We would like to thanks the reviewer for these positive comments on our work, and the careful reading. We will improve the manuscrit based on _all_ your minor comments, in particular regarding the presentation of the different objects introduced in our paper. We address in the following some specific comments that may merit discussions or clarifications.
>
> **Regarding injectivity**: We recall that there is no injective map from $\mathbb{R}^d \to \mathbb{R}$ for $d > 1$, and we emphasize that require only injectivity from the (potential) support to $\mathbb{R}$. It is true that asking for injectivity of a fine grid would have numerical consequences, but here we merely need it on the real support of the transport problem. We would like to add that this condition is satisfied almost surely in the linear case, and could be also satisfied with injective neural network [1]. Note that we require this property to prove that we obtain a metric, the object $d^\theta$ _still_ make sense without this hypothesis. In particular, we did not observe performance collapse due to noninjective $\phi$ map. From a numerical point of view, we simply rely on the order given by the sort algorithm used by Python that can be interpreted as a selection map from the set of minimizers.
>
> [1] Puthawala, Michael, et al. "Globally injective relu networks." _Journal of Machine Learning Research_ 23.105 (2022): 1-55.
>
> **Regarding the differentiability**: Thanks for this comment. You're absolutely right. Formally, we should work in the context of Bolte & Pauwels [2] with conservative Jacobians. Since this kind of notions is far away from the interest of the OT community of NeurIPS, we believe that it is better to keep a differentiable hypothesis to simplify the exposition. We will add a comment in the revision in the direction of a formal proof with conservative Jacobians (that are exactly design to tackle for instance ReLU functions).
>
> [2] Bolte, Jérôme, and Edouard Pauwels. "Conservative set valued fields, automatic differentiation, stochastic gradient methods and deep learning." _Mathematical Programming_ 188.1 (2021): 19-51.
>
> **Regarding practical benefits over SWGG**. You are right, we erroneously introduce ASWD as AWD in Sec. 2.2 which impairs understanding. This will be fixed in the next iteration of the paper.
> Regarding comparison between min-SWGG (optim) and DGSWP (linear) at equal timings (sorry, figures are not allowed), we get (note that the iteration for DGSWP (linear) is chosen here so as to lead to a timing lower or equal to that of the corresponding min-SWGG run):
>
> | Dataset | Method | Iteration | $\log_{10} W$ | Timing |
> |---|---|---|---|---|
> | Swiss Roll | min-SWGG (optim) | 500 | **-0.81** | 3.34 |
> | Swiss Roll | DGSWP (linear) | 173 | -0.49 | 3.34 |
> | Swiss Roll | min-SWGG (optim) | 1000 | **-0.89** | 6.67 |
> | Swiss Roll | DGSWP (linear) | 345 | -0.84 | 6.67|
> | Swiss Roll | min-SWGG (optim) | 1500 | -0.80 | 10.00 |
> | Swiss Roll | DGSWP (linear) | 516 | **-1.07** | 9.99 |
> | Swiss Roll | min-SWGG (optim) | 2000 | -0.79 | 13.33 |
> | Swiss Roll | DGSWP (linear) | 687 | **-1.28** | 13.32 |
> | Gaussian 2d | min-SWGG (optim) | 500 | **-1.42** | 3.40 |
> | Gaussian 2d | DGSWP (linear) | 172 |-0.40| 3.38 |
> | Gaussian 2d | min-SWGG (optim) | 1000 |**-2.01**| 6.81 |
> | Gaussian 2d | DGSWP (linear) | 342 |-1.12| 6.80 |
> | Gaussian 2d | min-SWGG (optim) | 1500 |**-1.96**| 10.25 |
> | Gaussian 2d | DGSWP (linear) | 514 |-1.75| 10.24 |
> | Gaussian 2d | min-SWGG (optim) | 2000 |-2.28| 13.67|
> | Gaussian 2d | DGSWP (linear) | 686 |**-2.46**| 13.66 |
>
> We will add such numbers in the next iteration of the paper and discuss the fact that min-SWGG can be more efficient early in the process and DGSWP (linear) becomes more accurate as running time grows.
>
> **Regarding the comparaison with Expected Sliced Wasserstein [35].** When considering ESW as an additional competitor, we get the results from Tables S.1 and S.2 below:
> For the dataset in Fig. 1, varying the temperature parameter $\tau$ for Expected Sliced Wasserstein, we get the following results in terms of transportation cost:
>
> _Table S.1_
>
> | Method | $\langle \pi, C\rangle$ cost |
> |---|---|
> | Exact OT | 13.50 |
> | min-SWGG | 22.22 |
> | Expected Sliced Wasserstein ($\tau=0$) | 32.28 |
> | Expected Sliced Wasserstein ($\tau=1$) | 21.30 |
> | Expected Sliced Wasserstein ($\tau=10$) | 20.09 |
> | DGSWP (NN) | 13.80 |
>
> For the gradient flow experiment in Fig. 4, we get the following performance after 2,000 iterations for the Swiss Roll dataset (note that results are consistent across all 4 datasets).
>
> _Table S.2_
>
> | Method | $\log_{10} W$ (at convergence) |
> |---|---|
> | min-SWGG (optim) | -0.79 |
> | ESW ($\tau=0$) | -0.07 |
> | ESW ($\tau=1$) | -0.08 |
> | ESW ($\tau=10$) | -0.13 |
> | DGSWP (NN) | **-5.83** |
>
> In other words, in the experimental setup of Fig. 1 and Fig. 4, ESW is consistently outperformed by both min-SWGG and DGSWP in terms of transportation cost and convergence of the corresponding gradient flows (varying the hyper-parameter from $\tau=0$ to $\tau=10$).
> Based on these initial comparisons on toy datasets, we have decided not to include ESW in the comparison for larger-scale problems such as the CFM experiment.
>
> **Regarding function evaluations.** In the CFM context, at generation time, given a model for $v(x, t)$, different solvers can be used to integrate the path from $x_0$ to $x_1$. It is common in the CFM litterature to evaluate the complexity of the generation steps as the number of calls to $v$ required for the integration [13, 34, 54].
> Note that in all our experiments, $v$ is implemented using a UNet architecture, which is the same across methods. In practice, between I-CFM, OT-CFM and DGSWP-CFM, the only difference lies in the training procedure that is used to train $v$.
> As a consequence, NFE, which is the number of times the model $v$ is called is the adequate measure of complexity for the generation step.
>
> **Regarding local minima w.r.t $\theta$.** You are absolutely right, DGSWP is non-convex and it converges towards a local minima. Even with simple scenario with 4 samples can lead to non-convex objective function. To assess the impact of the initialization, we perform several runs for each experiments. We notice that, in practice, results are similar across runs, even for the CFM experiment with is in dimension $d> 3000$.
>
> **Regarding positive curvature.** It is indeed a very interesting question. DGSWP necessitates to define a $\phi$ function that projects the sample $\in \mathbb{R}^d \to \mathbb{R}$, and uses a sliced mechanism to obtain the permutations (i.e. it suffices to sort the samples according to their projection). It is not straightforward to adapt DGSWP to the context of spherical slicing: for instance, [4] projects on a circle (and not on a line), which necessitates to employ the SW distance on the circle; [5] uses stereographic projections that could be used but is not naturally rotationaly invariant. We believe that some adaptations of DGSWP should allows it to be adapted in this context but we leave it for future work.
> Regarding positive curvature spaces, and again relying the formulation of [6] to define manifold feature maps $\mathcal{M} \to \mathbb{R}$, one could naturally use DGSWP to compare SPD matrices (see [7] for an example with sliced-Wasserstein).
>
> [4] Bonet, C., Berg, P., Courty, N., Septier, F., Drumetz, L., & Pham, M. T. Spherical Sliced-Wasserstein. ICLR.
>
> [5] Tran, H., Bai, Y., Kothapalli, A., Shahbazi, A., Liu, X., Diaz_Martin, R. P., & Kolouri, S. (2024). Stereographic Spherical Sliced Wasserstein Distances. ICML.
>
> [6] Katsman, I., Chen, E., Holalkere, S., Asch, A., Lou, A., Lim, S. N., & De Sa, C. M. (2023). Riemannian residual neural networks. NeurIPS.
>
> [7] Bonet, C., Malézieux, B., Rakotomamonjy, A., Drumetz, L., Moreau, T., Kowalski, M., & Courty, N. (2023). Sliced-Wasserstein on symmetric positive definite matrices for M/EEG signals. ICML

---

> > ### Comment · Reviewer_kzcn · 2025-08-03
> > **the authors addressed my concerns**
> >
> > the authors addressed my concerns, as usual they are biased concerning positive curvature and in particular spherically sliced OT and a better look into the literature would be good,
> > nevertheless,  since this is not the focus of the paper I will raised my score to 5

---

> > > ### Author Response · Authors · 2025-08-04
> > > **Post-rebuttal**
> > >
> > > We thank the reviewer for their thoughtful feedback and for raising their score.

---

### Note · Authors · 2025-08-12

We thank all reviewers for the constructive discussions, which helped clarify both theoretical and experimental aspects of our work. In the final version, we will: (i) improve the presentation of introduced objects and clarify our contributions, including the role of injectivity; (ii) correct the AWD/ASWD typo; (iii) add runtime complexity estimates and qualitative runtime comparisons; (iv) include additional numerical results (multi-run CIFAR-10/CFM, ESW) and theoretical complexity analysis; and (v) expand comments on differentiability with conservative Jacobians and variance reduction.

---

### Decision · Program_Chairs · 2025-09-17

**Decision:**

Accept (poster)

**Comment:**

The authors proposed an approach to calculate a sliced wasserstein plan. They generalized a standard approach based on a linear projection to more general functions including neural networks. Also, for efficient usage of SGD they proposed a trick to calculate gradients using Stein’s lema.

As a strengths of the paper I would highlight
- The approach works well compared to similar approaches of such kind,
- The approach is generalized  to non-linear projections via neural networks and to manifold-valued data
- The paper is well written

As a weaknesses I would highlight
- To some extent the approach is a combination of existing ideas (GSW and min-GSW)

The main reasons for the accept decision are
- Sufficiently novel generalization of existing algorithm
- Good experimental verification
- Sufficiently clearly written text

During the discussion with the reviewers several issues were raised that should be addressed in the final version of the paper. Here is the list of the main issues
- Improve the text and clarify the main contributions, provide information about computational performance
 (reviewer kzcn, reviewer WERw)
- More extensive experiments with error bars and comparison with expected sliced OT (reviewer Q6uB)
- Compare the performance of other slicing methods on the flow matching experiment (reviewer 8PTq)